# REAL DEEP RESEARCH FOR AI AND ROBOTICS

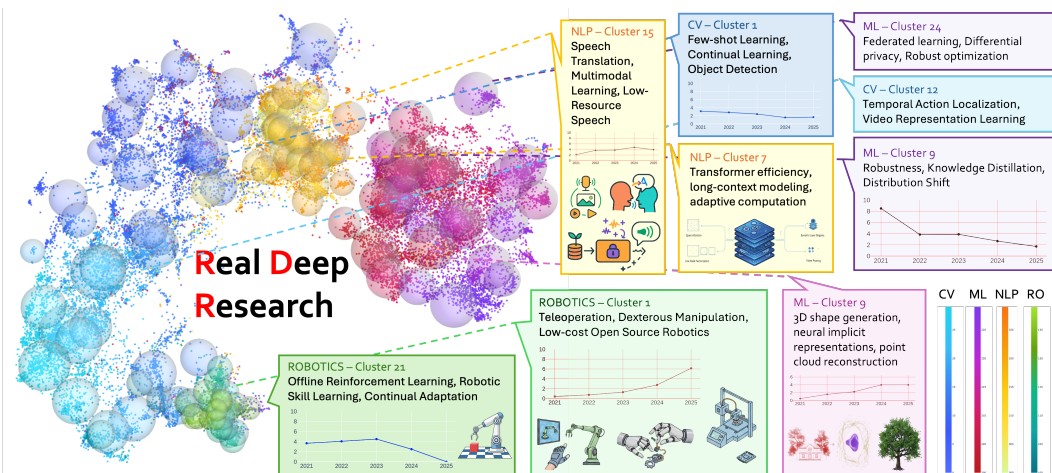

Figure 1: Real Deep Research enables: (1) generating surveys for specific research focuses or perspectives; (2) analyzing topic trends over time; (3) mapping interdisciplinary research landscapes; and (4) retrieving high-impact papers relevant to a given topic. (Each dot represents a paper, and each sphere denotes a topic cluster. The cluster keywords and trend information are automatically generated by **RDR**)

## ABSTRACT

With the rapid growth of research in AI and robotics—now producing over 10,000 papers annually—it has become increasingly difficult for researchers to stay up to date. Fast-evolving trends, *the rise of interdisciplinary work*, and *the need to explore domains beyond one's expertise* all contribute to this challenge. To address these issues, we propose a generalizable pipeline capable of systematically analyzing any research area: identifying emerging trends, uncovering cross-domain opportunities, and offering concrete starting points for new inquiry. In this work, we present **Real Deep Research** (RDR)—a comprehensive framework applied to the domains of AI and robotics, with a particular focus on foundation models and robotics advancements. We also briefly extend our analysis to other areas of science. The main paper details the construction of the RDR pipeline, while the appendix provides extensive results across each analyzed topic. We hope this work could shed lights on researchers who works in the filed of AI and beyond.

## 1 INTRODUCTION

The fields of AI and robotics have experienced exponential growth in recent years, while researchers continue to face the constraint of limited time and attention. This work is motivated by the authors' need to efficiently survey research areas, stay up to date with rapidly evolving trends, identify promising interdisciplinary opportunities, and quickly familiarize themselves with the latest developments on a given topic.

In response to this need, we develop a systematic analysis tool designed to help users quickly navigate and adapt to any research area or topic. We begin by applying our approach to the fields of AI and robotics, conducting an in-depth analysis with a focus on foundation models and robotics research. To broaden our exploration and uncover emerging areas of interest, we also extend our

analysis to natural sciences and formal sciences, offering a glimpse into recent developments beyond our core domains.

While our intentions are well-founded, it is important to acknowledge existing efforts in this space. On one hand, there are high-quality survey papers written by domain experts (3, 11); on the other, a few recent works have explored automated research pipelines (1, 18). Expert-written surveys offer depth and accuracy but require significant manual effort and cannot easily adapt to the fast-paced evolution of research. Meanwhile, current automated approaches often lack domain-specific knowledge and expert insight, limiting their usefulness and relevance to researchers. Our work aims to bridge this gap by combining systematic automation with meaningful, expert-informed analysis.

Therefore, in addition to building an effective pipeline for Real Deep Research, our goal is to make the tool robust and insightful enough to support top-tier researchers in tracking emerging trends and engaging with unfamiliar research areas. A key focus of our work is interdisciplinary exploration—helping researchers identify underexplored intersections between fields that present promising opportunities for cross-domain collaboration.

As shown in Fig. 1, the visualization displays individual papers, clustered research topics, and their corresponding trends. At a glance, it becomes clear that areas such as teleoperation, dexterous manipulation, and open-source robotics are emerging as promising directions, whereas traditional reinforcement learning appears to be declining in momentum. As researchers in the robotics field, we find that these trend insights align well with our domain knowledge and provide valuable guidance for identifying impactful research opportunities. We summarize the key contributions of this paper as follows:

1. We propose the **Real Deep Research (RDR)** pipeline, a systematic framework for exploring and analyzing any research area in depth.
2. Leveraging domain expertise, we deliver high-quality survey outputs in the fields of AI and robotics, providing valuable insights for researchers and practitioners.
3. We quantitatively evaluate the RDR pipeline and demonstrate its advantages over existing commercial large language model tools within the targeted research domains.

## 2 RELATED WORK

**Surveys of Foundation Models.** In recent years, a number of survey studies have systematically reviewed foundation models across different domains (3, 11, 7, 24, 53, 41), including natural language processing (51, 6), computer vision (24, 47), graph learning (41), and robotics (46, 44, 27, 42). However, these surveys require extensive manual effort and become outdated quickly due to the rapid progress of foundation models. Unlike such static surveys, our goal is to design a framework that can automatically analyze thousands of papers and provide an always up to date understanding of different research areas.

**LLMs in Scientific Research.** Large language models (LLMs) have been applied across various stages of scientific research (37, 25, 30, 35), including idea generation (39, 2), coding (43, 28), paper reviewing (21, 25), and predicting experimental results (29, 26). Among these stages, literature analysis plays a central role, involving tasks such as paper retrieval, clustering, and topic trend analysis. However, traditional literature search tools such as Google Scholar rely mainly on lexical matching and struggle with tasks that require deeper semantic reasoning. This has motivated researchers to leverage LLMs for literature analysis (1, 18, 10, 36). For example, SciLitLLM (18) employs supervised learning to build a specialized LLM for scientific literature understanding; PaSa (10) uses reinforcement learning with synthetic data to train an LLM agent that can answer complex scholarly queries. Unlike prior work that focuses mainly on research question answering, our approach targets a broader and systematic understanding of entire research areas. We highlight not only semantic reasoning over large collections of papers but also automatic analysis of research trends, offering researchers a transparent and evidence-based view of the literature.

**Knowledge Organization and Discovery.** It has been shown that LLMs are capable of clustering documents (38, 49) and uncovering latent topics (33, 17). For example, Knowledge Navigator (14) combines LLMs with clustering techniques to organize and structure documents for scientific literature search; SciTopic (17) enhances LLMs in identifying topic structures by refining document embeddings. Beyond knowledge organization, recent research (15, 16, 8) also studies the trend of

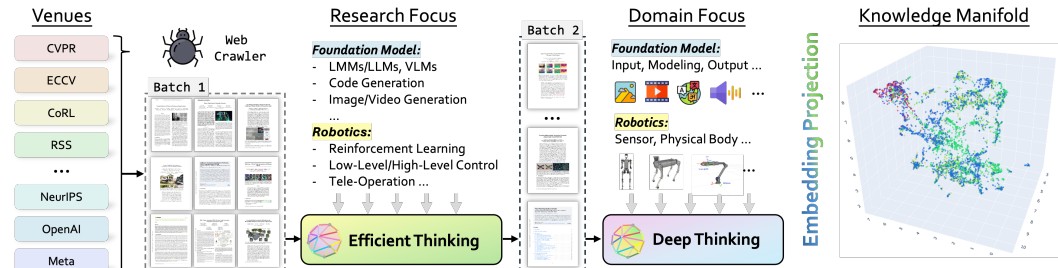

Figure 2: Pipeline of the proposed method on filtering and projecting thousands of papers to the embedding space for future analysis.

high-impact research topics. Our work introduces a novel approach by leveraging the reasoning capabilities of LLMs and the embedding representations of foundation models, which leads to more accurate and semantic knowledge organization. Built on this knowledge structure, our framework enables analysis of past and future research trends and supports inspection of connections between topics, providing valuable insights into scientific directions.

## 3 METHOD

In the Methods section, we focus specifically on the domains of foundation models and robotics to provide a comprehensive overview of how we conduct Real Deep Research using expert knowledge. As illustrated in Fig. 2, the embedding-based analysis pipeline consists of four main components:(1) Data Preparation (Sec. 3.1), (2) Content Reasoning (Sec. 3.2), (3) Content Projection (also in Sec. 3.2), and (4) Embedding Analysis (Sec. 3.4). This pipeline is powered by a suite of large language and multimodal models (LLMs/LMMs) for content extraction and reasoning, and is designed to be generalizable for the automated analysis of other research domains in the future. The following sections introduce each component in detail.

### 3.1 DATA PREPARATION.

**Selection.** To systematically investigate the integration of foundation models and robotics at scale, we focus on emerging trends and research priorities in both academia and industry. To capture the latest developments, we review recent publications from leading conferences in computer vision, robotics, and machine learning. Specifically, we collect papers via web crawling from top conference venues (CVPR, ECCV, ICCV, CoRL, RSS, ICRA, NeurIPS, etc.) as well as from industry research platforms (Nvidia, Meta, and OpenAI, etc.). This curated corpus comprehensively overviews the research contents in foundation models and robotics, highlighting key technical advancements, existing challenges, and future research directions. Specifically, we collect paper titles, authors, abstracts, and PDF links directly from conference and company websites. Then, we apply an area filtering process on paper titles and abstracts using an efficient LLM with a predefined set of criteria to ensure relevance to this study.

**Area Filtering.** We define the collected paper set as $\mathbf{P}$, while it generally fall under the broad area of vision, language, machine learning, and robotics, it is not guaranteed that each paper directly aligns with the specific focus of our work, such as foundation models ($\mathbf{D}_f$) and robotics ($\mathbf{D}_r$). To address this, we introduce *Area Filtering*—a step that leverages an efficient LLM with curated prompts—to identify papers relevant to our research scope. To ensure a correct filtering, we first define the scope of foundation models and robotics, clarifying technical boundaries between domains. Below are the prompts that we designed for our research focus:

```
Foundation Model Definition:  ''Research involving deep learning models
(especially transformer-based) trained on large amounts of data and
capable of fitting generalized factual realities.  These models typically
serve as versatile backbones for a variety of downstream tasks across
multiple domains.''
Key Indicators:
- Large Multimodal Models (LMM)
- Large Language Models (LLM) ...
```

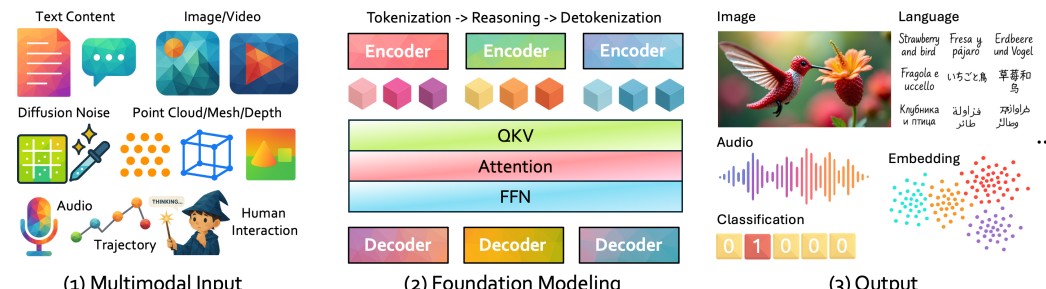

Figure 3: Perspective analysis of foundation model research, which primarily includes (1) Input, (2) Modeling, (3) Output, etc., shown in the figure.

```
Robotics Definition:  ''Research involving hardware systems equipped
with input sensors and mechanical kinematics capable of producing joint
movements.  These systems are controlled by learning-based algorithms
that facilitate automatic or robust mappings from sensory inputs to
actuator outputs.''
Key Indicators:
- Reinforcement Learning in robotic contexts
- Imitation Learning for physical systems ...
```

After filtering using an efficient LLM, the resulting set of papers ($\mathbf{P}'$) belongs to either the foundation model domain ($\mathbf{D}_f$), the robotics domain ($\mathbf{D}_r$), or both. Formally we write $\mathbf{P}' = \{p \mid p \in \mathbf{D}_f \cup \mathbf{D}_r\}$.

## 3.2 CONTENT REASONING.

Given the filtered papers $\mathbf{P}'$ in the domains of foundation models and robotics, an in-depth analysis is required to narrow the position of each paper. Guided by domain experts in foundation models and robotics, we define perspectives that align with established domain structures, emerging trends, and evolving knowledge. Beyond predefined perspectives, our pipeline supports future user-defined perspectives, allowing adaptation to new research questions. In the following paragraphs, we will depict the general structure, trends, and knowledge of the foundation model and robotics in preparation for analyzing the research works under $\mathbf{P}'$.

**Foundation Model.** A foundation model's development are systematically analyzed through five fundamental perspectives in this work: Input ($\mathbf{I}$), Modeling ($\mathbf{M}$), Output ($\mathbf{O}$), Objective ($\mathbf{W}$), and Learning Recipe ($\mathbf{R}$). We have shown some main perspectives examples in Fig. 3. This structured representation facilitates a comprehensive analysis of the foundation model. Below is the formal writing for the procedure:

$$\mathcal{D}_f^{P'} = \bigcup_{p \in \mathbf{P}'} F(p), \quad F(p) = \text{LLM}(p \mid \mathbf{I}, \mathbf{M}, \mathbf{O}, \mathbf{W}, \mathbf{R}),$$

where LLM represents the large multimodal model, and $\mathcal{D}_f^{P'}$ denotes the perspective projection of the given papers in $\mathbf{P}'$, focused on foundation model research. In the following paragraphs, we provide a formal definition of each perspective:

*Input ($\mathbf{I}$). The input processing for a foundation model generally involves raw data and a tokenization procedure. Standard input sources include images, videos, audio, LiDAR, etc., with tokenization performed through transformations and neural networks.*

*Modeling ($\mathbf{M}$). With input settled for a foundation model, the modeling part is responsible for extracting critical knowledge from the input, reasoning, and decoding to the output space. It is the critical procedure to transfer input knowledge to output.*

*Output ($\mathbf{O}$). The task determines the decoding space according to the input and modeling, this is the final step to decode the latent representation to the output used for loss computation or the final interaction.*

*Objective ($\mathbf{W}$). To fit a foundation model with the corresponding input, and output, the given model architecture is constrained by the learning objective, this fits the model distribution in alignment with the transformation given the task(s).*

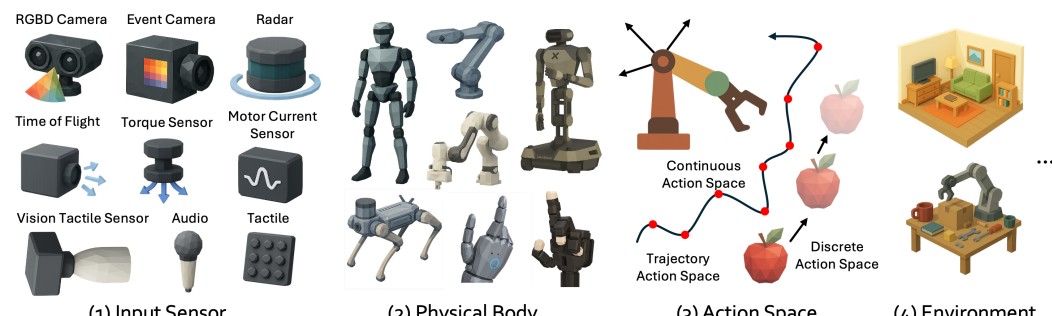

Figure 4: Perspective analysis of robotics research, which primarily includes (1) Input, (2) Modeling, (3) Output, etc., shown in the figure.

*Recipe (**R**). The recipe is used as the cookbook on how to tune the model weight with input, output, and objective. It controls the training stage, convergence speed, and updated parameters.*

**Robotics.** For research work in robotics, the core perspective shifts to emphasize hardware and interaction within real-world environments. We define five key perspectives to map each paper within the broader landscape of robotic applications: Input Sensor (**S**), Physical Body (**B**), Joint Output (**J**), Action Space (**A**), and Environment (**E**). An example of core perspectives is illustrated in Fig. 4. These perspectives collectively define how robots perceive, act, and interact within the physical world. The procedure could be formally written as:

$$\mathcal{D}_r^{P'} = \bigcup_{p \in \mathbf{P'}} F(p), \quad F(p) = \text{LMM}(p \mid \mathbf{S}, \mathbf{B}, \mathbf{J}, \mathbf{A}, \mathbf{E}),$$

where $\mathcal{D}_r^{P'}$ represents the perspective projection of the given papers in $\mathbf{P'}$, providing a structured framework for analyzing robotics research. We show the concrete definition in the following:

*Input Sensor (**S**). Input sensors are hardware devices that measure physical quantities or environmental conditions and convert them into digital signals that can be processed by the robot's control system. They serve as the robot's interface with its environment.*

*Physical Body (**P**). A physical body in robotics refers to the mechanical structure and architecture that enables physical interaction with the environment. This physical manifestation determines how motor commands translate into real-world forces, movements, and environmental manipulations.*

*Action Space (**A**). The action space is the set of all permissible actions a robot can select in a given context, ranging from low-level joint commands to high-level behaviors (e.g., "walk" or "grasp"). Each chosen action is ultimately executed as a joint output, bridging decision-making to physical movement.*

*Joint Output (**J**). Joint output refers to the physical movement or configuration of a robot's joints resulting from executed motor commands. It translates control signals (e.g., torque or velocity) into mechanical motion, allowing the robot to directly interact with and manipulate its environment.*

*Environment (**E**). The environment encompasses the physical space where a robot operates, characterized by its spatial layout, structural features, and contextual elements (e.g., furniture, tools, obstacles) that shape the task-specific challenges and opportunities the robot encounters.*

Given the predefined perspective definition, we use the following prompt to extract each perspective from the paper:

```
Can you analyze the paper contents according to the following
perspectives: (1) Definition 1, (2) Definition 2, (3) Definition 3, ...
After analysis, please identify each of the perspectives in the paper,
and return the answer in the following format: {"perspective 1": plain
text, "perspective 2": plain text, "perspective 3": plain text, ...}
```

## 3.3 CONTENT PROJECTION.

Given the extracted contents from research papers guided by our defined perspective, we aim to project natural language descriptions into an informative latent space. This projection enables large-

scale analysis of current research in foundation models and robotics while revealing potential future research directions. Motivated by recent advancements in large language model-based embedding models, we **employ a pre-trained embedding foundation model G** to project $\mathcal{D}_r^{P'}$ (processed robotics papers' content) and $\mathcal{D}_f^{P'}$ (processed foundation model papers' content) from natural language space into a more abstract embedding space. The embedding model maps text into a high-dimensional vector space where semantically similar concepts occupy proximate regions.

We formally define this projection procedure as follows: For any text snippet $x \in \mathcal{D}$, its embedding is computed as: $v_x = \mathbf{G}(x) \in \mathbb{R}^d$. Our core assumption is that by projecting paper contents through this perspective-aware embedding process and analyzing them in the high-dimensional manifold, we can uncover meaningful patterns, research trends, and potential gaps in the literature through systematic visualization and clustering analysis.

### 3.4 EMBEDDING ANALYSIS.

The goal of embedding analysis is to structure the understanding of previously extracted embeddings. The pipeline for embedding analysis contains three components: (1) Clustering on the extracted embeddings and analyzing the main concept from each cluster. (2) Structured the concept for each cluster to formulate an informative table. (3) Given the structured understanding, we trace back to the reference papers.

**Clustering for Embeddings.** We first embed every paper to obtain its vector representation $V$ and partition the corpus into $k$ clusters. From each cluster, we then draw a random sample of 50 papers and feed their text to a reasoning-based model with the prompt:

```
Can you summarize the following contents into three distinct keywords:
Here is one example output:"keyphrase1, keyphrase2, keyphrase3".  The
output should be short and precise, with a single output for all
papers.
```

The model returns three compact key phrases that capture the cluster's core theme, giving every paper both a cluster label and an interpretable set of keywords for subsequent analysis.

**Structuring for Thoughts.** With clustered embeddings and their associated topic keywords in place, the next step is to generate a structured survey for the given research area. To accomplish this, we leverage the o3 language model, using the clustered keywords as prompts to guide the formulation of the final survey content. Incorporating the clustering results into the prompt ensures that the generated text remains grounded in the actual structure of the research landscape, enhancing both coherence and relevance. We use the following prompt to produce the final output:

```
  Those are summarized keywords for a number of science papers clustered
by abstract contents, however they are ambiguous, contents may overlap
between clusters, can you summarize the information in a more structured
way for audience with the following criteria:  ...
```

## 4 ANALYSIS

In this section, we conduct a comprehensive qualitative analysis of the conclusions drawn in this work from the following perspectives: (1) Embedding analysis for general research areas. (2) Embedding analysis within specific perspective. (3) Trend analysis of research focus over time. (4) Knowledge graph exploration across different research areas. (5) Retrieval examples based on embeddings. This pipeline will enable a researcher to dive into any research area, identify what to explore, and determine the specific papers to focus on.

**Embedding Analysis - General.** The output of the embedding analysis is a comprehensive survey tailored to the featured research domain. This survey is organized into major categories and sub-categories, each detailing the specific topics covered. Rather than generating the survey content via LLM, we leverage the clustering results from the embedding analysis to guide its structure and scope. Additionally, for each sub-topic, we include the most relevant citations to provide readers with direct references for further exploration. We have provide the full survey for Foundation Model, Robotics, Computer Vision, Natural language processing, and machine learning in Appendix. B.

| Cat. | Sub-category | What is covered | Typical examples | Cluster |
|---|---|---|---|---|
| **1. Perception & Mapping (34, 48, 52, 31, 9)** | | | | |
| | 1.1 Multimodal sensor fusion (20, 45, 4, 54, 40) | Fuse heterogeneous sensors for richer scene understanding | LiDAR–Camera Fusion; Radar–Camera Fusion; V2X Cooperative Perception ... | 0,6,7,8, 14,16 |
| | 1.2 3D reconstruct/occupancy (52, 12, 22, 23, 32) | Build dense or sparse geometric maps for localisation | 3-D SLAM & Reconstruction; 3-D Occupancy; Efficient 3-D Representation | 0,8,16 |
| | 1.3 BEV / top-view mapping (50, 45, 5, 13, 19) | Bird's-eye or top-down representations for planning | BEV Perception; V2X Collaborative Perception | 0,14,16 |
| ... ... | | | | |

**Embedding Analysis - Perspective.** After establishing a clear overview of the domain, we analyze it through a targeted *perspective* to expose structure and problem formulations. As introduced in the Methods section, our perspective analysis uses embedding-based clustering to organize works along a chosen axis. In this study we focus on foundation models and robotics, examining how each community formulates its problems. Below we illustrate the robotics case from the viewpoint of *action space*. This perspective-guided embedding analysis yields a deeper understanding of the domain and a high-level map of how researchers approach and solve its problems. We also provide the full perspective for foundation model and robotics in Appendix. C.

| Category | Sub-category | What is covered | Typical examples | Cluster |
|---|---|---|---|---|
| **1. Continuous Low-Level Actuation** | | | | |
| | 1.1 Joint-space commands | Direct numerical inputs to individual joints or actuators, bounded by hardware limits. | joint torques/positions/velocities; high-dimensional joint commands; bounded control inputs; finger-joint configs; parametrised joint trajectories | 0, 4, 6, 10, 12, 14, 18 |
| | 1.2 Vehicle / body dynamics commands | Low-level controls that change a mobile base, ground-vehicle or aerial body state. | steering angle; throttle / acceleration; braking; linear & angular velocity; body-rate thrust; speed/direction for locomotion; lane-keeping | 0, 1, 7, 10, 12, 13, 15 |
| ... ... | | | | |

**Trend Analysis.** Once we understand each domain and its key sub-perspectives, the next step is to assess topic momentum. Our trend analysis highlights which areas are accelerating and which have been thoroughly explored in recent years, giving a practical starting point when entering a new field. In robotics (see figure), the trajectories suggest that teleoperation, dexterous manipulation, and low-cost open-source robotics are currently rising, while traditional reinforcement learning and skill-based manipulation appear comparatively mature or show slowing activity. This will guide the researchers to smoothly enter a new field. We provide the full trend analysis in Appendix. D for Computer Vision, NLP, Robotics, and Machine Learning.

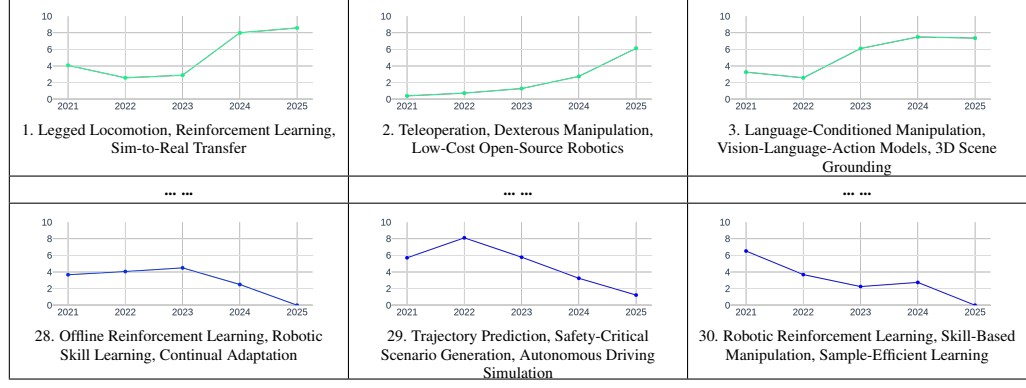

**Knowledge Graph.** Beyond identifying trending topics, a complementary path to new research directions is to surface cross-domain themes. In this work, we analyze intersections among computer vision, natural language processing, machine learning, and robotics. The left figure presents a cross-domain topology graph: colors denote domains; nodes (spheres) represent topics; edges indicate inter-domain connections; and endpoints mark domain-specific topics with no current cross links. This view reveals a mix of genuinely cross-domain areas and isolated, domain-specific topics—promising targets for future cross-pollination and inter-disciplinary exploration.

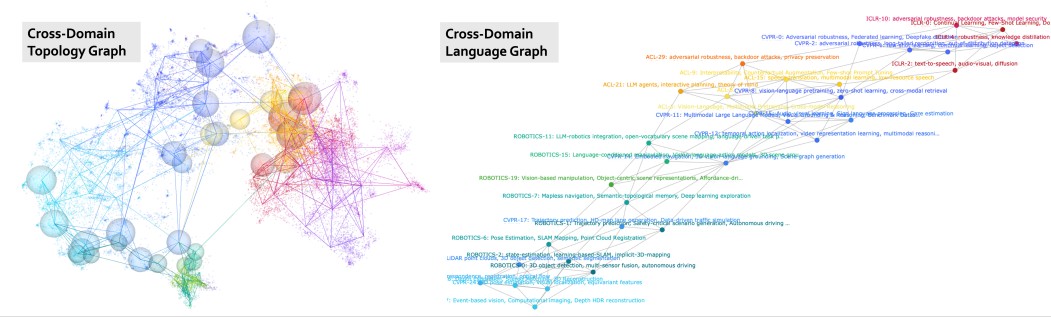

**Retrieval Examples.** Once a target research topic is identified, the next step is to pinpoint concrete entry points. We do this by leveraging the conference-level embeddings inferred earlier to run semantic searches and retrieve the most relevant literature. For example, after surveying robotics, we focus on dexterous manipulation and query the embedding index to surface closely related papers across venues. As shown in the table below, the returned papers align tightly with the query and exhibit meaningful community impact, as reflected by their venues, years, and citation counts.

| Paper | Year | Venue | Citations |
|---|---|---|---|
| **Query: dexterous manipulation generated data in 3D simulation and evaluated in real world.** | | | |
| Evaluating Real-World Robot Manipulation Policies in Simulation | 2024 | CoRL24 | 127 |
| Lessons from Learning to Spin "Pens" | 2024 | CoRL24 | 29 |
| General In-hand Object Rotation with Vision and Touch | 2023 | CoRL23 | 134 |
| Twisting Lids Off with Two Hands | 2024 | CoRL24 | 13 |
| DexterityGen: Foundation Controller for Unprecedented Dexterity | 2025 | RSS25 | 16 |

## 5 EXPERIMENT

We have presented a comprehensive qualitative analysis demonstrating how Real Deep Research supports deep dives into a chosen research focus. This section now details the dataset curated for our study and the implementation specifics required to realize the system. We then provide quantitative evaluations—both by benchmarking our survey against commercial research tools and by validating the effectiveness of the embeddings that underpin our approach.

**Dataset.** We curate our dataset from publicly available venues. Since the main paper centers on AI and robotics, we collect publications from the venues listed in Table 1. To align with our focus on foundation models and robotics, we further filter the corpus to include 4,424 foundation-model papers across all venues since 2024 and 1,186 robotics papers over the same period. These statistics indicate that, relative to the broader AI literature, robotics remains a smaller community—one that is well positioned for growth, especially through cross-domain research. We plan to expand the paper collection over time.

| Venue | Year | Area | Total |
|---|---|---|---|
| CVPR | 21-25 | *Computer Vision* | 11668 |
| CoRL | 21-24 | *Robotics* | 815 |
| RSS | 21-25 | *Robotics* | 575 |
| ICLR | 21-25 | *Machine Learning* | 9549 |
| ACL | 21-25 | *NLP* | 4556 |
| NeurIPS | 2024 | *Machine Learning* | 4240 |
| ECCV | 2024 | *Computer Vision* | 6166 |

Table 1: **Paper Distribution Analysis** across different venues, showing the total number of papers.

**Implementation Details.** We do not train any new networks in this work for generating the embedding or survey; instead, we rely on off-the-shelf models. For straightforward tasks—such as classifying research areas—we use the Doubao language model. For reasoning-intensive tasks and complex summarization, we employ the o3 model to achieve stronger performance. To extract text embeddings, we use nvidia/NV-Embed-v2.

**Survey Quality.** As demonstrated in Sec. 3, our analysis of a research area begins with a broad survey of the existing literature. The analysis pipeline we propose is designed to significantly reduce model hallucination and produce a comprehensive, high-quality survey for a given research direction.

| Model | Rank | General | | | | Foundation Model | | | Robotics | | |
|---|---|---|---|---|---|---|---|---|---|---|---|
| | | CV | NLP | ML | Robotics | Input | Modeling | Output | Sensor | Body | Action |
| GPT5 | 4.80 | 10.00 | 17.39 | 45.45 | 71.43 | 44.44 | 10.00 | 21.05 | 22.73 | 34.78 | 69.57 |
| GPT5-Thinking | 2.75 | **82.61** | 59.09 | 47.83 | 66.67 | 55.00 | **90.91** | 50.00 | 88.46 | 42.86 | 32.00 |
| GPT5-Research | 4.00 | 42.11 | 50.00 | 72.73 | 63.64 | 21.05 | 35.00 | 50.00 | 0.00 | 40.91 | 52.63 |
| Gemini | 4.80 | 35.00 | 40.00 | 15.38 | 0.00 | 13.64 | 54.17 | 45.83 | 31.25 | 45.00 | 26.32 |
| Gemini-Thinking | 3.35 | 63.64 | 50.00 | 56.25 | 37.50 | 65.22 | 45.45 | 41.67 | 55.56 | 56.52 | 34.78 |
| RDR (Ours) | **1.30** | 58.33 | **89.47** | **73.68** | **77.78** | **88.46** | 60.00 | **94.74** | **91.30** | **84.21** | **89.47** |

Table 2: Survey Quality Evaluation among RDR and commercial based methods. We evaluate the pairwise winning rate for each domain and perspective.

To evaluate the accuracy and quality of the generated surveys, we conducted a user study involving experienced researchers with domain expertise in robotics and foundation models. As a baseline, we prompted a commercial large language model using the following instruction: *"Act as an expert research analyst. Your task is to create a structured map of the research landscape for a given academic or industrial field. The output must be a single, valid JSON object that categorizes the field into its primary research areas and specific sub-topics. For the research area 'foundation model,' can you summarize the input perspective with the following definition: The input processing for a foundation model generally involves raw data and a tokenization procedure ..."*

To assess the quality of the generated surveys, we adopted a pairwise comparison methodology rather than asking evaluators to select a single best output. For each comparison, domain experts were presented with two survey outputs and asked to determine which one demonstrated superior quality and accuracy. This evaluation setup helps reduce cognitive load and bias, making the assessment more reliable by avoiding the need for evaluators to recall or rank multiple outputs simultaneously. In total, we collected 8 evaluation entries, each with 80 pairwise comparisons. To quantify performance, we computed the winning rate of each method within its respective domain.

As shown in Tab. 5, our method, Real Deep Research (RDR), achieves the highest overall performance with an average rank of 1.30, outperforming all baselines. RDR leads in key domains such as NLP (89.47), robotics (77.78), and foundation model output (94.74), and also shows strong performance in robotics subfields like sensor (91.30) and action (89.47). While GPT5-Thinking slightly outperforms in CV (82.61) and foundation model modeling (90.91), RDR consistently ranks at or near the top across nearly all categories.

**Embedding Quality.** Because much of our analysis relies on high-quality embeddings, we evaluate their effectiveness using a simple lin-

| Model | AG News | | | 20 News Groups | | |
|---|---|---|---|---|---|---|
| | ACC(↑) | NMI(↑) | ARI(↑) | ACC(↑) | NMI(↑) | ARI(↑) |
| LDA | 74.05 | 47.17 | 49.01 | 29.05 | 31.63 | 13.34 |
| NMF | 34.05 | 4.59 | 2.13 | 12.42 | 12.86 | 0.48 |
| ProdLDA | 80.93 | 56.51 | 60.91 | 37.42 | 45.67 | 23.89 |
| DecTM | 55.63 | 40.04 | 36.17 | 36.57 | 46.18 | 22.90 |
| ETM | 26.14 | 0.00 | 0.00 | 5.35 | 0.10 | 0.00 |
| NSTM | 26.14 | 0.01 | 0.00 | 16.92 | 17.02 | 2.34 |
| TSCTM | 79.63 | 53.91 | 55.89 | 40.60 | 44.06 | 15.71 |
| ECRTM | 78.69 | 54.05 | 54.88 | 25.70 | 31.00 | 12.26 |
| Bertopic | 35.93 | 12.88 | 7.03 | 29.78 | 28.57 | 11.58 |
| FASTopic | 83.48 | 59.10 | 62.48 | 51.65 | 56.32 | 39.49 |
| SciTopic* | 85.29 | 61.96 | 65.94 | 70.88 | 68.32 | 55.71 |
| RDR (Ours) | 84.86 | 61.66 | 65.24 | 52.91 | 56.57 | 39.96 |

Table 3: Unsupervised Clustering performance. * indicate using more labels.

ear probe trained on top of frozen representations—an approach that best reflects the intrinsic utility of the embeddings themselves. We follow the experimental protocol introduced in SciTopic (17), using the same unsupervised training and evaluation splits to ensure fair comparison. Unlike our method, SciTopic uses pseudo-labels during training, which introduces weak supervision; therefore, we gray out its entry in the results for clarity. As shown in Tab. 5, our method RDR achieves the best performance across both datasets, with an accuracy of 84.86 on AG News and 52.91 on 20 News Groups. RDR also leads in NMI (61.66 and 56.57) and ARI (65.24 and 39.96), outperforming all fully unsupervised baselines, and even surpassing the pseudo-supervised SciTopic model.

## 6 CONCLUSION

We began this work with the goal of tracking research trends and broadening our understanding across fields to support interdisciplinary exploration. The RDR pipeline has since become a valuable tool for the authors' ongoing research. We hope it inspires future meta-research efforts toward LLMs that can one day conduct research autonomously.

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

# A APPENDIX

This appendix presents the complete results of our Real Deep Research (RDR) analysis across a wide range of domains. We include detailed domain-level surveys (e.g., AI, robotics, computer vision, natural language processing), perspective-based breakdowns (e.g., input/output modeling in foundation models, sensor/action perspectives in robotics), and trend analyses to track the evolution of research focus over time. These results collectively offer a structured and insightful view of the research landscape, serving as a valuable reference for both new and experienced researchers.

**Use of Large Language Model.** In paper writing, we use LLM to fix grammars and find part of related work. And we clearly indicate in which part we use LLM to do generate contents and extract features.

# B  DOMAIN SURVEY

| Cat. | Sub-category | What is covered | Typical examples | Cluster |
|---|---|---|---|---|
| **1. Model modalities & representations** | | | | |
| | 1.1 Vision–Language | Foundation models that jointly process images/video and natural language. | Vision-Language Models; Vision-Language Robotic FMs | 0, 3 |
| | 1.2 Multimodal (>=3) | Architectures/objectives agnostic to the exact mix of modalities. | Multimodal Foundation Models; Multimodal LLMs; Multimodal Large Language Models | 1, 4, 5, 6 |
| | 1.3 Open-vocabulary grounding | Linking free-form text to modality-specific regions or anchors. | Open-Vocabulary Grounding | 1 |
| | 1.4 3D/4D & video reps | Learned neural representations for 3-D/4-D scenes and video. | 3D & Multimodal Representation; Diffusion-based 3D/4D Generation; 3D & Video Synthesis | 0, 7, 9 |
| | 1.5 Neural scene encoding | Representations enabling view-consistent reconstruction. | Multi-view Consistent Reconstruction; Gaussian Splatting; NeRF Representations | 9 |
| **2. Generative & diffusion techniques** | | | | |
| | 2.1 Core diffusion modelling | Diffusion processes used as the primary generative backbone. | Diffusion Generative Modeling | 7 |
| | 2.2 Control & personalisation | Steering diffusion outputs with prompts, adapters or user profiles. | Controllable Diffusion Personalization; Controllable Efficient Sampling | 7, 11 |
| | 2.3 Robot policy via diffusion | Using diffusion to learn control policies for robots or manipulators. | Diffusion-Based Policy Learning | 3 |
| | 2.4 Editing & post-generation | Applying diffusion to edit or refine existing content. | Diffusion-based Generative Editing | 4 |
| | 2.5 Efficiency & distillation | Speed-ups and compact student models for diffusion. | Diffusion Model Acceleration; Efficient Sampling & Distillation | 8 |
| **3. Training & adaptation strategies** | | | | |
| | 3.1 Self-/pre-training paradigms | Large-scale unsupervised or weakly-supervised pre-training methods. | Elastic Self-Supervised Pre-training | 6 |
| | 3.2 Prompt/adapter learning | Lightweight modulation of frozen backbones via prompts or adapters. | Prompt/Adapter Tuning; Prompt/Adapter Learning; Parameter-Efficient Prompt Tuning | 0, 1, 5 |
| | 3.3 Param-efficient finetune | LoRA/adapters that tune only a small slice of parameters. | Parameter-Efficient Fine-Tuning; Adapter-Efficient Fine-Tuning | 10, 11 |
| | 3.4 Compression & inference efficiency | Sparsity, low-rank factorisation and runtime acceleration. | Sparse/Low-Rank Model Compression; Efficient Transformer Inference | 10 |
| **4. Safety, alignment & ethics** | | | | |
| | 4.1 Safety alignment | Aligning model behaviour with human or policy constraints. | LLM Safety Alignment; Alignment & Safety | 2, 6 |
| | 4.2 Bias & harm mitigation | Detecting and reducing social or representational bias. | Safety & Bias Mitigation | 11 |
| | 4.3 Preference optimisation | Fine-tuning with human preference or RLHF-style signals. | Preference-Optimized Fine-Tuning | 2 |
| **5. Embodied interaction & robotics** | | | | |
| | 5.1 Robotic foundation models | General-purpose models for perception and control on robots. | Vision-Language Robotic Foundation Models | 3 |
| | 5.2 Embodied agents | Agents acting in simulated or real environments with multimodal inputs. | Embodied Vision-Language Agents | 4 |
| | 5.3 Intended manipulation | Grounding natural-language instructions into robot actions. | Multimodal Instruction-Guided Manipulation | 3 |
| **6. Reasoning & agent systems** | | | | |
| | 6.1 Multi-agent reasoning | Coordinated planning or dialogue among several learned agents. | Multi-Agent Reasoning | 2 |
| **7. Generalisation & robustness** | | | | |
| | 7.1 Domain robustness | Techniques to maintain performance under domain shift. | Domain-Robust Generalization | 5 |

Table 4: Domain Survey for Foundation Model.

| Cat. | Sub-category | What is covered | Typical examples | Cluster |
|---|---|---|---|---|
| **1. Perception & Mapping** | | | | |
| | 1.1 Multimodal sensor fusion | Fuse heterogeneous sensors for richer scene understanding | LiDAR–Camera Fusion; Radar–Camera Fusion; V2X Cooperative Perception ... | 0,6,7,8,14,16 |
| | 1.2 3D reconstruct/occupancy | Build dense or sparse geometric maps for localisation | 3-D SLAM & Reconstruction; 3-D Occupancy; Efficient 3-D Representation | 0,8,16 |
| | 1.3 BEV / top-view mapping | Bird's-eye or top-down representations for planning | BEV Perception; V2X Collaborative Perception | 0,14,16 |
| **2. Manipulation & Grasping** | | | | |
| | 2.1 Dexterous grasping | Multi-finger in-hand manipulation | Dexterous Robotic Grasping; Dexterous Grasp & Manipulation | 11,12 |
| | 2.2 Generalist manipulation | Single policy handles diverse objects/tasks | Generalist Robotic Manipulation; Robotic Manipulation; Humanoid Manipulation | 3,4,9 |
| | 2.3 Tactile-vision fusion | Combine touch and vision for reactive grasps | Multimodal Tactile-Vision Learning | 11 |
| **3. Locomotion & Navigation** | | | | |
| | 3.1 Legged locomotion control | Whole-body control and adaptation on uneven terrain | Legged Robot Locomotion; Learning-Based Control & Adaptation | 17 |
| | 3.2 Embodied VL navigation | Language-directed navigation with active mapping | Embodied Vision-Language Navigation; Active 3-D Mapping & Planning | 7,19 |
| **4. Planning & Control** | | | | |
| | 4.1 Language/hierarchical planning | Translate language or high-level goals into executable skills | Language-Guided Planning & Control; Hierarchical Skill Planning & Adaptation | 2,18 |
| | 4.2 Diffusion/Transformer policies | Trajectory generation with generative sequence models | Diffusion Policies; Diffusion/Transformer Policy Models; Generative Diffusion Models | 1,9,14 |
| **5. Robot Learning & Adaptation** | | | | |
| | 5.1 RL & imitation | Learn skills from rewards, demonstrations or offline data | Robot Reinforcement Learning; Imitation Learning Policies; Sample-Efficient RL ... | 3,9,15 |
| | 5.2 Sim to real & continual adaptation | Transfer and improve policies across domains over time | Continual Sim-to-Real Adaptation; Sim-to-Real Transfer; Self-Supervised Distillation/Adaptation | 0,4,15,16 |
| | 5.3 Multitask / generalisable policies | Single policy generalises across many tasks and embodiments | Multitask Generalisable Robotics | 1 |
| **6. Autonomous Driving** | | | | |
| | 6.1 Motion forecasting, perception & simulation | Forecast traffic actors, all-weather perception, long-tail scenario simulation | Motion Forecasting; Trajectory Prediction; Driving Perception; Scenario Simulation | 5,6,8,13,14 |
| **7. Simulation & World Models** | | | | |
| | 7.1 Generative world models | Learn latent physics/world models for planning or RL | Generative World Models | 12 |
| | 7.2 Self-supervised simulation | Expand synthetic experience using self-supervised signals | Self-Supervised Generative Simulation | 5 |
| **8. Embodied Language Robotics** | | | | |
| | 8.1 LLM-driven robotics | Use large language models for zero-shot policy/reasoning | LLM-Driven Robotics; LLM-Enhanced Driving; LLM-Driven Zero-Shot Planning | 2,13,19 |
| | 8.2 Vision-language control | Pair vision with text to drive low-level actions | Vision-Language Robotic Control; Hierarchical Skill Planning & Adaptation | 18 |
| | 8.3 Open-vocabulary mapping | Build scene maps labelled with free-form language | Open-Vocabulary Scene Mapping | 19 |
| **9. Safety & Robustness** | | | | |
| | 9.1 Safety-aware planning | Explicit risk reasoning during motion generation | Safety-Aware Motion Planning | 10 |
| | 9.2 Runtime monitoring | Detect and mitigate failures on-the-fly | Failure Detection & Runtime Monitoring | 18 |
| | 9.3 Robust control | Improve stability against disturbances and uncertainties | Safety & Robustness (Locomotion) | 17 |
| **10. Multi-Robot & Human Collaboration** | | | | |
| | 10.1 Multi-agent collaboration | Plan and act with other robots or humans in the loop | Multi-Agent / Human-Robot Collaboration | 10 |

Table 5: Domain Survey for Robotics.

| Category | Sub-category | What is covered | Typical examples | Cluster |
|---|---|---|---|---|
| **1. Robust & Generalizable Learning** | | | | |
| | 1.1 Adversarial / OOD Robustness | Defending against or detecting malicious, anomalous or out-of-distribution inputs. | adversarial robustness; deepfake detection; anomaly detection; out-of-distribution detection | 0,2,5 |
| | 1.2 Domain Adaptation & Generalization | Transferring models across different domains, devices or persons without performance drop. | domain adaptation; domain generalization; test-time adaptation; person re-identification | 3,6 |
| | 1.3 Low-Data Learning | Learning reliably from scarce, imbalanced or continually arriving data. | few-shot learning; continual learning; long-tailed recognition; federated learning | 0,1,2 |
| **2. Representation & Model Efficiency** | | | | |
| | 2.1 Representation Learning & Distillation | Un/semisupervised learning and distillation techniques that build informative, explainable features. | self-supervised learning; semi-supervised segmentation; pseudo-label consistency; representation learning; knowledge distillation; explainability | 4,5,7,10 |
| | 2.2 Efficient Architectures | Designing compact, hardware-friendly or automatically searched neural networks. | efficient vision transformers; neural architecture compression; sparse NAS; quantized NAS | 9 |
| **3. Generative Modeling & Editing** | | | | |
| | 3.1 2D Generative Imaging | Synthesising or editing images/videos with controllable appearance or compression. | generative adversarial networks; image inpainting; image translation; neural style transfer; diffusion-based image/video generation; controllable generative editing; neural compression | 18,21,23 |
| | 3.2 3D Neural Rendering & Scene Generation | Generating or reconstructing 3-D scenes via implicit or explicit neural representations. | neural radiance fields; 3D scene generation; 3D Gaussian splatting; dynamic scene reconstruction; neural rendering | 25,28,29 |
| **4. 2D Perception & Enhancement** | | | | |
| | 4.1 Detection & Segmentation | Locating objects or semantic regions in images/videos. | object detection; semantic segmentation; few-shot detection | 1,10,16 |
| | 4.2 Tracking & Motion Estimation | Following objects or estimating pixel correspondences over time. | object tracking; correspondence; registration; optical flow; UAV surveillance | 13,20 |
| | 4.3 Matting & Transparency | Separating foreground layers or transparency effects in images/videos. | image/video matting; trimap guidance; mask guidance; transformer-based matting models | 19 |
| | 4.4 Restoration & Enhancement | Improving quality of degraded images/videos or reconstructing HDR. | image/video restoration; diffusion models for restoration; HDR reconstruction | 21,27 |
| **5. 3D Perception & Geometry** | | | | |
| | 5.1 Depth & Reconstruction | Estimating depth or reconstructing 3-D structure from images. | depth estimation; stereo matching; 3D reconstruction | 26 |
| | 5.2 LiDAR & 3D Detection | Understanding point clouds for object detection and semantic segmentation. | LiDAR point clouds; 3D object detection; 3D semantic segmentation | 16 |
| | 5.3 Pose & Localization | Estimating 6-D object poses or localizing cameras in space. | 6D pose estimation; visual localization; equivariant features | 24 |
| **6. Video & Temporal Understanding** | | | | |
| | 6.1 Temporal Action & Video Reasoning | Recognising and localising actions or reasoning over temporal video cues. | temporal action localization; video representation learning; multimodal reasoning | 12 |
| **7. Multimodal & Vision-Language Systems** | | | | |
| | 7.1 Vision-Language Pretraining & Retrieval | Learning cross-modal representations for zero-shot tasks or retrieval. | vision-language pretraining; zero-shot learning; cross-modal retrieval | 8 |
| | 7.2 Multimodal Large Models & Grounding | Large models that jointly reason over vision and language with grounding. | multimodal large language models; visual grounding; visual reasoning; benchmark datasets; 3D vision-language grounding; scene graph generation | 11,14 |
| | 7.3 Audio / Sign / Gaze Multimodality | Integrating audio, sign language or gaze with vision tasks. | audio-visual learning; sign language processing; gaze estimation | 15 |
| **8. Human-Centric Understanding & Animation** | | | | |
| | 8.1 Pose & Interaction | Estimating human body pose and modelling human-object interactions. | 3D human pose estimation; human-object interaction; transformer-based motion generation | 22 |
| | 8.2 Avatars & Animation | Building and animating realistic 3-D human avatars. | 3D human avatars; pose-driven animation; neural rendering of humans | 28 |
| **9. Embodied & Autonomous Systems** | | | | |
| | 9.1 Embodied Navigation & Mapping | Perception and planning for agents navigating 3-D environments. | embodied navigation; HD-map generation; lane generation | 14,17 |
| | 9.2 Trajectory Prediction & Traffic Simulation | Forecasting future paths and simulating realistic traffic participants. | trajectory prediction; data-driven traffic simulation | 17 |

Table 6: Domain Survey for Computer Vision.

| Category | Sub-category | What is covered | Typical examples | Cluster |
|---|---|---|---|---|
| **1. Information Extraction** | | | | |
| | 1.1 Entity & Relation Extraction | Automatic detection of named entities plus the semantic relations or events connecting them. | Named Entity Recognition, Relation Extraction, Event Extraction | 0 |
| **2. Text Generation & Summarization** | | | | |
| | 2.1 Summarization & Keyphrase Generation | Producing concise summaries or keyphrases from longer documents. | Summarization, Keyphrase, Evaluation | 12 |
| | 2.2 Controllable & Stylistic Generation | Generating text under user-specified style or attribute constraints. | Style transfer, controllable text generation, representations | 27 |
| **3. Dialogue & Conversational Systems** | | | | |
| | 3.1 Task-oriented Dialogue | Dialogue systems that track state and generate responses to accomplish user goals. | Dialogue systems, Response generation, Dialogue state tracking | 14 |
| | 3.2 Empathetic & Safe Dialogue | Handling empathy, hate speech and multimodal cues in conversations. | HateSpeechDetection, EmpatheticDialogue, ... | 25 |
| **4. Multilingual & Cross-lingual NLP** | | | | |
| | 4.1 Multilingual Modeling & Transfer | Building models that operate across many (often low-resource) languages and transfer knowledge between them. | low-resource languages, multilingual language models, cross-lingual transfer | 16 |
| | 4.2 Multilingual Machine Translation | Neural translation among multiple language pairs, often using shared or distilled models. | Neural machine, Knowledge distill, Multilingual modeling | 19 |
| | 4.3 Multimodal Low-Resource Speech | Speech translation/recognition when data are scarce or involve multiple modalities. | speech translation, multimodal learning, low-resource speech | 15 |
| **5. Knowledge & Reasoning** | | | | |
| | 5.1 Knowledge Graph Reasoning | Embedding and temporal/causal reasoning over structured knowledge graphs. | knowledge graph embedding, event causality reasoning, temporal knowledge reasoning | 1 |
| | 5.2 Mathematical & Chain-of-Thought Reasoning | Using large language models for step-by-step logical or mathematical reasoning. | Large language models, Mathematical reasoning, Chain-of-thought prompting | 10 |
| | 5.3 Compositional & Syntactic Generalization | Probing or improving models to generalize compositionally or parse syntax. | syntactic parsing, compositional generalization, lan. model probing | 3 |
| **6. Retrieval & Question Answering** | | | | |
| | 6.1 Dense Retrieval & RAG | Learning dense vector search for open-domain QA and retrieval-augmented generation. | Dense retrieval, open-domain question answering, retrieval-augmented generation | 4 |
| | 6.2 Table & Structured QA / Generation | Mapping natural language to SQL, answering table queries or generating text from structured data. | Text-to-SQL, Table Question Answering, Data-to-Text Generation | 2 |
| **7. Evaluation, Alignment & Editing** | | | | |
| | 7.1 LLM Evaluation & Human Alignment | Designing metrics and feedback loops to align large language models with human intent. | LLM evaluation, alignment methods, human feedback | 23 |
| | 7.2 Hallucination, Calibration & Knowledge Editing | Detecting/mitigating false outputs and editing or calibrating model knowledge. | hallucination, knowledge editing, calibration | 11 |
| | 7.3 Evaluation Metrics & Data Augmentation | Developing metrics and synthetic data (incl. figurative language) to assess or improve models. | Evaluation metrics, Data augmentation, Figurative language | 13 |
| **8. Model Training Paradigms & Efficiency** | | | | |
| | 8.1 Continual, In-context & Instruction Tuning | Allowing models to learn new tasks or follow instructions without full retraining. | continual learning, instruction tuning, in-context learning | 17 |
| | 8.2 Parameter-Efficient & Compressed Models | Reducing training/inference cost via adapters, pruning or lightweight fine-tuning. | parameter-efficient fine-tuning, model compression for LLMs | 20 |
| | 8.3 Transformer Efficiency & Long-Context Modeling | Architectural or computational methods to scale transformers to longer contexts efficiently. | Transformer efficiency, long-context modeling, adaptive computation | 7 |
| | 8.4 Sentence & Multilingual Representation Learning | Contrastive or related methods to build versatile sentence embeddings across languages. | multilingual representation learning, sentence embeddings, contrastive learning | 6 |
| **9. Safety, Bias & Robustness** | | | | |
| | 9.1 Social Bias & Fairness | Measuring and mitigating demographic or social biases in NLP systems. | social bias, debiasing, fairness evaluation | 26 |
| | 9.2 Misinformation & Fact Verification | Detecting false claims, AI-generated text or aligning model values with truthfulness. | fact verification, misinformation detection, evidence retrieval, fake detection, value alignment | 18,28 |
| | 9.3 Security & Privacy Robustness | Protecting models against adversarial, backdoor or privacy attacks. | adversarial robustness, backdoor attacks, privacy preservation | 29 |
| **10. Agents & Interactive Reasoning** | | | | |
| | 10.1 LLM-based Agents & Planning | Using large language models as autonomous agents capable of interactive planning and theory-of-mind reasoning. | LLM agents, interactive planning, theory of mind | 21 |
| **11. Code Intelligence** | | | | |
| | 11.1 Code Generation & Benchmarks | Generating executable code and evaluating models on coding tasks. | code generation, large language models, benchmark evaluation | 24 |

Table 7: Domain Survey for Natural Language Processing.

| Category | Sub-category | What is covered | Typical examples | Cluster |
|---|---|---|---|---|
| **1. Generative Modelling & Media Synthesis** | | | | |
| | 1.1 Image / Video Generation & Editing | Models that create or edit 2-D or temporal visual content via generative techniques. | Generative modeling; Image synthesis/editing; Diffusion-based methods; Optimal Transport; Diffusion Models | 1, 3 |
| | 1.2 3-D Object & Molecule Generation | Generating 3-D shapes or molecular structures using geometry-aware or equivariant models. | 3D shape generation; neural implicit representations; point cloud reconstruction; equivariant GNNs; 3D molecular generation; drug discovery | 9, 16 |
| | 1.3 Audio & Speech Synthesis | Producing speech or audio from text or multimodal cues, via diffusion models. | text-to-speech; audio-visual; diffusion | 2 |
| **2. Representation & Transfer Learning** | | | | |
| | 2.1 Continual / Few-Shot / Domain Adaptation | Adapting models continually, with few examples, or across shifting domains. | Continual Learning; Few-Shot Learning; Domain Adaptation | 0 |
| | 2.2 Self-, Contrastive & Retrieval-Augmented Learning | Building rich representations via self/contrastive learning or external retrieval augmentation. | Self-supervised Learning; contrastive learning; disentangled representations; clustering; language-modeling; retrieval-augmentation; representation-learning | 5, 6, 15 |
| | 2.3 Parameter-Efficient Transfer | Adapting large transformers with minimal new parameters and compute. | Efficient-transformer-architectures; parameter-efficient-fine-tuning; multilingual-adaptation | 7 |
| **3. Robustness, Security & Privacy** | | | | |
| | 3.1 Adversarial & Backdoor Robustness | Defending against adversarial or backdoor manipulations and distribution shifts. | adversarial robustness; backdoor attacks; robustness; knowledge distillation; distribution shift | 10, 4 |
| | 3.2 Uncertainty & Interpretability | Quantifying model confidence and explaining predictions. | uncertainty estimation; conformal prediction; model interpretability | 14 |
| | 3.3 Privacy & Machine Unlearning | Ensuring data privacy and enabling deletion or secure distributed learning. | differential privacy; machine unlearning; federated learning; robust optimization | 19, 24 |
| **4. Model Efficiency & Compression** | | | | |
| | 4.1 Pruning, Quantization & Embedding Compression | Compressing networks by pruning, quantizing or embedding reduction for efficient deployment. | Network pruning; Sparse_Network_Pruning; Low-precision quantization; Embedding_Compression; Efficient architecture search; Recommendation_Systems | 8, 21 |
| **5. Geometric & Graph Learning** | | | | |
| | 5.1 Equivariant / Geometric Deep Networks | Networks that respect group symmetries to learn geometric or physical structures. | equivariant neural networks; group symmetry; geometric deep learning | 12 |
| | 5.2 Graph Neural Network Theory | Theoretical properties, expressivity and robustness of Graph Neural Networks. | Graph Neural Networks; Expressivity; Robustness | 20 |
| **6. Optimization & Theory** | | | | |
| | 6.1 Non-convex & Stochastic Optimization | Algorithms and analysis for nonconvex optimization with stochastic gradients. | Nonconvex optimization; Stochastic gradient methods; Convergence analysis | 18 |
| | 6.2 Neural Network Theory & Neuroscience Inspiration | Theoretical studies and neuro-inspired modeling of recurrent nets. | recurrent neural networks; neuroscience-inspired modeling; theoretical analysis | 17 |
| **7. Reinforcement Learning & Embodied Intelligence** | | | | |
| | 7.1 Core & Offline Reinforcement Learning | Improving sample efficiency and offline policy learning in RL. | Reinforcement Learning; Offline Learning; Sample Efficiency | 29 |
| | 7.2 Multi-Agent & Dialogue RL | Learning cooperation, competition or dialogue among multiple agents. | multi-agent reinforcement learning; bandit algorithms; game-theoretic learning; dialogue systems; multi-agent collaboration; reinforcement learning | 28, 26 |
| | 7.3 Embodied AI & Robotics | Training embodied agents and robots via differentiable simulation and manipulation tasks. | Embodied AI; Robotic manipulation; Differentiable simulation | 27 |
| **8. Multimodal Perception & Reasoning** | | | | |
| | 8.1 Vision-Language & Knowledge Reasoning | Joint reasoning across vision and language plus knowledge graphs. | vision-language reasoning; knowledge graph learning; compositional generalization | 11 |
| | 8.2 Video Understanding & 3-D Perception | Temporal and 3-D understanding of videos and dynamic scenes. | Video understanding; Temporal modeling; 3D perception | 13 |
| **9. Scientific & Symbolic Machine Learning** | | | | |
| | 9.1 Physics & Differential Equation-guided Learning | Learning operators governed by physical laws and differential equations. | Neural differential equations; Physics-informed operator learning; Spatiotemporal forecasting | 23 |
| | 9.2 Program Synthesis & Automated Reasoning | Automatically generating code or formal proofs from specifications. | Program synthesis; Code generation; Theorem proving | 22 |
| | 9.3 Combinatorial, Causal & Bayesian Optimization | Optimization over discrete structures, causal questions or Bayesian objectives. | Combinatorial optimization; Causal inference; Bayesian optimization | 25 |

Table 8: Domain Survey for Machine Learning.

| Category | Sub-category | What is covered | Typical examples | Cluster |
|---|---|---|---|---|
| **1. Life Sciences & Biomedicine** | | | | |
| | 1.1 Immuno-oncology & Metabolic Signalling | Immune mechanisms in cancer and metabolic cues that modulate them | T-cell immunity; tumour microenvironment; metabolic signalling | 0 |
| | 1.2 Cancer Genomics & Epigenetics | Genetic and epigenetic alterations driving oncogenesis and therapy response | Cancer; DNA repair; epigenetics | 1 |
| | 1.3 Infectious Disease & Microbiome Interactions | Host–pathogen dynamics and microbiome ecology shaping antimicrobial strategies | Host–pathogen interactions; antimicrobial therapeutics; microbiome dynamics | 2 |
| | 1.4 Neuro-immune Metabolism & Aging | Crosstalk between immune system, metabolism and brain across aging | Neuroimmunology; metabolism; aging | 3 |
| | 1.5 Genome Editing & Microbial/Plant Immunity | Engineering genomes and decoding microbial | plant defence mechanisms & CRISPR-based genome editing; bacterial anti-phage defence; plant immune signalling | 4 |
| | 1.6 Protein & RNA Engineering | Designing proteins and regulating chromatin/RNA to control cell function | Protein design; chromatin regulation; RNA processing | 5 |
| | 1.7 Neural Epigenetics & Disorders | Epigenetic regulation of neural plasticity and neuropsychiatric disease | Neural circuit plasticity; epigenetic regulation; neuropsychiatric disorders | 6 |
| | 1.8 Population & Single-Cell Genomics | Sequencing-based mapping of genetic variation at population & cellular resolution | Genome sequencing; population genetics; single-cell transcriptomics | 7 |
| | 1.9 Connectomics & Behaviour | Structural mapping of neural circuits to explain behaviour | Connectomics; neural circuit mapping; behaviour | 8 |
| | 1.10 Evolutionary Genomics & Paleobiology | Reconstructing evolutionary history using ancient DNA and fossils | Paleogenomics; prehistoric migrations; fossil record | 9 |
| **2. Chemistry & Materials Science** | | | | |
| | 2.1 Catalysis & Green Synthesis | Catalytic and synthetic routes for sustainable chemical production | Advanced catalysis; sustainable chemistry; synthetic methodologies | 10 |
| | 2.2 Functional Materials for Energy & Electronics | Multifunctional materials for energy storage and flexible devices | Advanced materials; energy storage; flexible electronics | 12 |
| | 2.3 Perovskite Solar Technologies | High-efficiency perovskite and tandem photovoltaics with interface engineering | Perovskite photovoltaics; tandem solar cells; interface passivation | 13 |
| | 2.4 Integrated Photonics & Optoelectronic Integration | Integration of perovskites and 2D semiconductors into photonic platforms | Integrated photonics; perovskite optoelectronics; 2D semiconductor integration | 14 |
| **3. Physics & Quantum Technology** | | | | |
| | 3.1 Quantum Materials | Emergent quantum phases in topological and moiré systems, incl. unconventional superconductivity | Topological quantum matter; moiré heterostructures; unconventional superconductivity | 16 |
| | 3.2 Quantum Computing Hardware & Networks | Scalable, fault-tolerant quantum processors and quantum communication links | Fault-tolerant quantum computing; scalable qubit hardware; quantum networking | 17 |
| **4. Earth & Environmental Science** | | | | |
| | 4.1 Climate Change & Ecosystem Impacts | How climate change alters ecosystems and the global environment | Climate-change; ecosystem-impacts; global-environment | 15 |
| **5. Astronomy & Astrophysics** | | | | |
| | 5.1 Exoplanetary Science with JWST | Characterising exoplanet atmospheres and interiors using JWST observations | Exoplanet atmospheres; planetary interiors; JWST observations | 18 |
| | 5.2 Early-Universe & Black-Hole Astronomy | Galaxy formation and supermassive black holes in the early Universe probed with JWST | JWST; early-Universe galaxies; supermassive black holes | 19 |
| **6. Computer Science & Artificial Intelligence** | | | | |
| | 6.1 Foundation & Trustworthy AI | Large foundation models, applied AI and methods ensuring fairness & reliability | Foundation models; applied artificial intelligence; fairness and reliability | 11 |

Table 9: Domain Survey for Natural related Topics.

| Category | Sub-category | What is covered | Typical examples | Cluster |
|---|---|---|---|---|
| **1. Earth & Environmental Sciences** | | | | |
| | 1.1 Climate & Ecosystem Dynamics | Interactions among climate change, carbon cycling and biodiversity, including conservation responses | Biodiversity loss; Conservation strategies; Climate change impacts; Climate change; Carbon cycle; Environmental impacts | 0,1 |
| | 1.2 Geophysical Processes | Physical processes shaping Earth's solid and cryospheric systems | Earthquakes; Volcanism; Ice dynamics | 2 |
| **2. Space Science** | | | | |
| | 2.1 Stellar & Space-Plasma Physics | Physics of stars, solar activity and the interstellar medium | Compact objects; Interstellar medium; Solar activity | 9 |
| **3. Biological Sciences** | | | | |
| | 3.1 Evolutionary Genomics | Genetic mechanisms driving adaptation and speciation | Evolutionary genomics; adaptation; speciation | 3 |
| | 3.2 Molecular & Cellular Regulation | Molecular signaling and structural mechanisms governing development and genome integrity | Hormone signaling; Immune defense; Developmental regulation; Genome stability; Chromosome segregation; Cryo-EM structural biology | 4,5 |
| | 3.3 Neurobiology & Systems Neuroscience | Gene-to-circuit bases of neural function, plasticity and behaviour | Neuroscience; Gene regulation; Single-cell; neural circuits; synaptic plasticity; behavior | 6,8 |
| | 3.4 Immunity, Infection & Therapy | Host defence mechanisms and engineered immunotherapies against pathogens and cancer | Antiphage immunity; CRISPR systems; Antibiotic discovery; Immunoregulation; Metabolic signaling; Cancer therapy; Infectious disease; Immunotherapy; Molecular engineering | 7,11,13 |
| | 3.5 Synthetic & Computational Biology | Design of biomolecules and biological systems using AI and synthetic methods | protein-design; deep-learning; synthetic-biology | 12 |
| **4. Materials & Chemical Sciences** | | | | |
| | 4.1 Catalysis & Chemical Transformations | Selective catalytic methods for constructing organic molecules | Catalytic organic synthesis; Radical-mediated transformations; Selective C–H functionalization | 14 |
| | 4.2 Advanced Functional Materials | Smart, biointegrated and nanostructured materials with tailored properties | Smart materials; Biointegrated electronics; Soft robotics; Nanostructured materials; Energy storage; Functional properties | 15,17 |
| | 4.3 Energy Conversion & Separation Materials | Materials enabling electrochemical, thermal and membrane-based energy technologies | Electrocatalysis; Porous framework materials; Membrane separations; Perovskite photovoltaics; Thermoelectric devices; Radiative cooling | 16,18 |
| **5. Physics & Quantum Technologies** | | | | |
| | 5.1 Quantum Materials & Information | Exotic quantum phases and their application to information processing | Topological quantum phases; Quantum information processing; Strongly correlated matter | 19 |
| **6. Computational & Social Systems Science** | | | | |
| | 6.1 Information Dynamics & Society | Computational study of information spread and persuasion in sociotechnical systems | misinformation propagation; democratic polarization; AI-mediated persuasion | 10 |

Table 10: Structured overview of clustered science research areas.

| Category | Sub-category | What is covered | Typical examples | Cluster |
|---|---|---|---|---|
| **1. Pharmacogenomics & Genetics-Guided Therapy** | | | | |
| | 1.1 Cytochrome P450 Genotype-Driven Therapy | Links CYP450 genetic variants to drug exposure and response for individualised dosing. | CYP2C19 pharmacogenetics, star-allele variability, precision antithrombotic therapy, antidepressant pharmacogenetics | 1,3,4 |
| | 1.2 Transporter Pharmacogenetics | Examines genetic variation in drug transporters and its impact on safety and efficacy. | SLCO1B1 variants, statin-associated muscle symptoms | 6 |
| | 1.3 PGx Implementation & Economic Evaluation | Assesses clinical decision support, workflow integration, and the cost-effectiveness of pharmacogenomics. | pharmacogenomics implementation, clinical decision support, cost-effectiveness | 11 |
| | 1.4 Oncology / High-Risk Therapy PGx | Uses germline variants to predict toxicity and guide dosing of high-risk or anticancer drugs. | DPYD variants, TPMT variants, NUDT15 variants, chemotherapy toxicity prediction, pharmacogenetic-guided dosing | 21 |
| **2. Quantitative Pharmacology & Model-Informed Drug Development** | | | | |
| | 2.1 Population PK & Dose Optimisation | Applies population PK and exposure–response models to refine dosing in diverse patients. | population pharmacokinetics, precision dosing, anticoagulants, exposure–response modelling, oncology real-world evidence | 2,17,25,29 |
| | 2.2 Mechanistic PBPK & Special Populations | Uses physiologically based PK models to predict drug disposition in paediatrics, the CNS, pregnancy, and other special populations. | paediatric PBPK modelling, CNS drug delivery, maternal–infant pharmacology, anti-infective therapy | 7,19 |
| | 2.3 Exposure–Response for Biologic / Cell Therapies | Characterises PK/PD and dose–response of biologics and cell-based therapies. | PK/PD modelling, haematological therapies, biologic PK/PD, immunomodulatory therapies, T-cell engagers | 15,26,28 |
| | 2.4 Machine-Learning-Assisted Precision Dosing | Integrates machine learning with PK models and real-world factors to individualise therapy. | machine learning, precision pharmacokinetic modelling, transplant immunosuppressant dosing, gut microbiota influence, model-informed drug development | 18,20 |
| **3. Drug Metabolism, Transport & Interaction Science** | | | | |
| | 3.1 Enzyme-Mediated DDIs & Prediction | Investigates cytochrome P450 interactions and uses models to forecast clinical risk. | cytochrome P450, PBPK modelling, QT prolongation | 14,5 |
| | 3.2 Transporter-Mediated DDIs & Biomarkers | Studies renal and hepatic transporters and endogenous probes to detect interaction liability. | renal transporters, hepatic transporters, endogenous biomarkers of transporter activity | 13,24,27 |
| | 3.3 Clinical DDIs & Risk Management | Documents real-world interaction scenarios and strategies to mitigate adverse outcomes. | opioid overdose management, nirmatrelvir/ritonavir interactions, EHR-based pharmacovigilance | 0,8 |
| **4. Regulatory Science & Evidence Generation** | | | | |
| | 4.1 Trial Diversity & Health Equity | Promotes representative enrolment and equitable access in clinical research. | clinical trial diversity, health equity, regulatory initiatives | 10 |
| | 4.2 Real-World Evidence & External Controls | Leverages observational data and synthetic controls to inform regulatory decisions. | real-world evidence, external control trials, regulatory frameworks | 9 |
| | 4.3 Drug-Lifecycle Oversight & Lag Analysis | Evaluates approval timelines, post-marketing requirements, and regulatory performance. | regulatory science, drug lag, post-marketing studies | 16 |
| | 4.4 Biomarker / Rare Disease / Biosimilar Qualification | Establishes evidentiary standards for biomarkers, orphan products, and biosimilars. | rare-disease drug development, biomarker qualification, biosimilar development, PK/PD biomarkers, regulatory strategies | 22,23 |
| **5. Clinical Therapeutics & Outcomes Research** | | | | |
| | 5.1 Cardio-Renal & Metabolic Outcomes | Assesses the long-term efficacy and safety of metabolic therapies on cardiovascular and renal endpoints. | SGLT2 inhibitors, cardiovascular–renal outcomes, antidiabetic drug safety | 12 |

Table 11: Domain Survey for Science related Survey.

# C  PERSPECTIVE SURVEY

| Category | Sub-category | What is covered | Typical examples | Cluster |
|---|---|---|---|---|
| **1. Textual inputs** | | | | |
| | 1.1 Large-scale tokenized corpora | Massive general-domain text for LM pre-training | Web pages; Wikipedia; books; C4; Pile; WikiText; OpenWebText; SlimPajama | 11 |
| | 1.2 Prompt & interaction data | User/system prompts and model replies gathered for alignment, RLHF or robustness | Prompts/questions; model responses; preference/reward labels; adversarial triggers; long-context demonstrations | 0, 2 |
| | 1.3 Problem statements with context | Natural-language tasks paired with explicit structured knowledge or code/data schemas | NL problem + knowledge graph/database schema/code stub; reasoning traces or step-by-step solutions | 14 |
| **2. Visual inputs (images)** | | | | |
| | 2.1 Raw images | Canonical labelled/unlabelled images after basic augmentation | ImageNet, CIFAR, COCO photos; medical scans | 19 |
| | 2.2 Cued images | Images supplied with auxiliary spatial/sensor cues | Low-light or blurry photos + masks; camera poses; depth/event data; points/boxes | 17 |
| | 2.3 Patch or region tokens | Visual patches embedded as tokens for transformer processing | ViT/MAE patches from images or single video frames | 3 |
| **3. Video & motion inputs** | | | | |
| | 3.1 Video streams with motion cues | Time-ordered frames plus motion/semantic signals | Video frames; optical flow; 3-D pose; segmentation masks; aligned audio track | 13 |
| **4. 3-D & spatial inputs** | | | | |
| | 4.1 Geometry & depth representations | Explicit 3-D or depth data for spatial reasoning | Point clouds; RGB-D images; TSDF/voxel grids; meshes; camera extrinsics; semantic labels | 1 |
| **5. Multimodal token sequences** | | | | |
| | 5.1 Cross-modal token bags | Tokens from diverse modalities embedded with positional info | Text, audio, vision, graphs, biology tokens with position vectors | 12, 18 |
| | 5.2 Encoder-fused tokens | Tokens from separate encoders concatenated into one sequence | CLIP/ViT image tokens + BERT/LLaMA text tokens | 15 |
| | 5.3 Normalized latent embeddings | Modality-specific encoders map data into a shared latent space (may include placeholders) | Text, images, video, audio all $\rightarrow$ joint embeddings (missing modalities allowed) | 4 |
| **6. Generative-model conditioning** | | | | |
| | 6.1 Diffusion noise schedule | Noisy latent sample $x_t$, timestep token $t$, optional class/text/geometry conditioning | $x_t + z$; timestep $t$; class label; pose map; depth; edges | 16 |
| | 6.2 Auxiliary generation cues | User-supplied hints steering image generation or editing | Reference image; mask; depth; pose; layout; bounding boxes | 10 |
| **7. Task-oriented multimodal inputs** | | | | |
| | 7.1 Visual observations + NL prompts | Perception frames paired with a natural-language task or edit instruction | Screenshot + "click the red button"; video frame + "highlight the pedestrian" | 6 |
| | 7.2 Image-text pairs with cues | Captioned/questioned images often carrying region annotations | Image + caption; VQA triplets; bounding-box / mask annotations | 7 |
| | 7.3 Embodied-agent context | Agent perception, proprioception | history combined with a goal description & RGB-D stream; past actions; goal text ("navigate to the chair") | 8 |
| **8. Sequential & trajectory inputs** | | | | |
| | 8.1 Offline state–action trajectories | Logged sequences for offline RL or behaviour cloning | Time-series control signals; graphs; 3-D skeleton poses; human preference labels | 9 |
| **9. Inverse-problem observations** | | | | |
| | 9.1 Corrupted measurements with ground truth | Raw measurements transformed by known operators, paired with target outputs | MRI $k$-space + mask; blurred $\rightarrow$ sharp image pairs; noisy sensor data | 5 |

Table 12: Structured summary of input types used in foundation-model papers.

| Category | Sub-category | What is covered | Typical examples | Cluster |
|---|---|---|---|---|
| **1. Representation & Architecture** | | | | |
| | 1.1 Token & latent representation | Mapping raw data to discrete/continuous tokens or latents | Latent representation learning; token/patch embedding; visual-token projection . . . | 0, 14, 17, 18, 19 |
| | 1.2 Attention & Transformer variants | Architectural changes that make attention cheaper or deeper | Sparse/low-rank attention; spatiotemporal attention; positional scaling . . . | 2, 10, 11, 14, 17, 18, 19 |
| | 1.3 Mixture-of-Experts & routing | Dynamic selection of expert blocks or routes | Modular MoE; dynamic routing; MoE token routing . . . | 0, 11, 15, 17, 19 |
| **2. Generative Paradigms** | | | | |
| | 2.1 Diffusion & score-based generation | Noise-to-data generative flows | UNet diffusion; latent diffusion; guided conditional sampling . . . | 2, 3, 4, 5, 10, 13, 18 |
| | 2.2 Energy-based & control formulations | Sampling by minimising energy or solving control processes | Energy-based score learning; optimal-control SDE; solver-accelerated inversion . . . | 13 |
| | 2.3 Probabilistic & masked inference | Non-diffusion probabilistic decoders | Probabilistic generative inference; masked auto-encoding; next-token prediction . . . | 0, 17, 18, 19 |
| **3. Multimodal Alignment & Fusion** | | | | |
| | 3.1 Encoders → shared latent space | Separate encoders project each modality into a common space | Modality-specific encoders; projection layers; frozen CLIP backbone . . . | 1, 6, 12, 18, 19 |
| | 3.2 Cross-attention fusion & conditioning | Mechanisms for interaction between modalities | Cross-attention fusion; prompt cross-attention; multimodal concatenation . . . | 2, 4, 6, 10, 12, 14, 18, 19 |
| | 3.3 Vision/Video-language alignment | Aligning paired modalities in latent space | Video-language alignment; contrastive alignment; generative self-supervision . . . | 1, 8, 12, 14, 18, 19 |
| **4. Adaptation & Efficiency** | | | | |
| | 4.1 Parameter-efficient adaptation | Updating only a small subset of weights or added modules | LoRA/adapters; low-rank tuning; modular fusion . . . | 1, 2, 4, 10, 12, 15, 16, 19 |
| | 4.2 Prompting & modular extensions | Steering frozen backbones with prompts or plug-ins | Prompt conditioning; chain-of-thought prompting; tool invocation . . . | 1, 6, 7, 9, 17, 19 |
| | 4.3 Compression & efficient training | Reducing compute, memory or training cost | Quantisation; pruning-distillation; communication-efficient sharding . . . | 2, 5, 10, 15, 16, 19 |
| **5. Reasoning & Interaction** | | | | |
| | 5.1 Chain-of-thought & tool reasoning | Explicit reasoning traces or calls to external tools | Chain-of-thought reasoning; retrieval-augmented reasoning; self-refinement . . . | 6, 7, 8, 9 |
| | 5.2 RL & preference modeling | Reinforcement or preference-based optimisation | Preference-conditioned policies; RLHF alignment; optimal-transport RL . . . | 3, 9 |
| | 5.3 Multi-agent / planner loops | Multiple interacting agents or explicit planner loops | Multi-agent collaboration; Planner-Actor-Corrector-Verifier loop . . . | 7, 8 |
| **6. Robustness & Domain Shift** | | | | |
| | 6.1 Uncertainty & robust optimisation | Estimating confidence and resisting adversarial inputs | Uncertainty quantification; adaptive memory; adversarial robustness . . . | 0, 9, 16 |
| | 6.2 Domain adaptation & model editing | Adapting or editing knowledge post-training | Targeted model editing; synthetic-data adaptation; knowledge probing . . . | 4, 9, 16, 19 |

Table 13: Structured summary of modeling techniques used in foundation-model papers.

| Category | Sub-category | What is covered | Typical examples | Cluster |
|---|---|---|---|---|
| **1. Language-centric outputs** | | | | |
| | 1.1 Token probabilities & sequences | Autoregressive LMs: token logits or generated text | next-token probability distributions; generated token sequences | 2 |
| | 1.2 Aligned LLM responses | Instruction-tuned completions for reasoning, safety, long-context | helpful-harmless-truthful responses; safe refusals; reasoning-enhanced; hallucination-reduced | 3, 15 |
| | 1.3 Reasoning traces & answers | Chain-of-thought steps plus final decoded result | chain-of-thought reasoning; intermediate outputs; final answers | 0 |
| | 1.4 Visually-grounded NL outputs | Language grounded in image/video content | captions; visual-QA; reasoning with bounding boxes, masks | 9 |
| **2. Generative visual & multimodal outputs** | | | | |
| | 2.1 Photorealistic images | High-fidelity images from text or prompts | photorealistic; identity-preserving; context-coherent images; text-conditioned images | 1, 18 |
| | 2.2 Video / motion generation | Consistent video or 3D motion from text | temporally-consistent video; 3-D motion generation/editing | 5 |
| | 2.3 3D scenes & assets | Meshes, NeRFs, Gaussian fields for rendering/editing | meshes; point clouds; NeRF; editable scene/asset generation | 11 |
| | 2.4 Multimodal reconstructions | Images, video, audio decoded from latents | reconstructed/generated multi-modal data | 19 |
| | 2.5 Diffusion samples & noise | Reverse-diffusion outputs with noise estimates | generated or reconstructed samples... plus noise/score estimates | 7 |
| **3. Predictive & structured outputs** | | | | |
| | 3.1 Classification scores | Class labels, probabilities, or logits | class labels / probabilities / logits | 4, 13 |
| | 3.2 Localization & segmentation | Masks, boxes, poses, or captions pinpointing content | segmentation masks; bounding boxes; 3-D localization/pose; textual grounding/captions | 16, 13 |
| | 3.3 Structured artefacts | Graphs, coordinates, flows, molecules, causal terms | graphs; poses; flows; molecular/crystal structures; uncertainties; causal/physical parameters | 14 |
| | 3.4 Downstream embeddings | Transformed features for later task use | transformed feature embeddings; reconstructed/generative outputs | 13 |
| | 3.5 Control & planning | Predicted actions, plans, or trajectories | action sequences & control commands; task-grounded plans | 6 |
| **4. Evaluation, improvement & efficiency outputs** | | | | |
| | 4.1 Metrics & benchmarks | Accuracy, bias, uncertainty, safety, etc. | accuracy; F1; ROC-AUC; Elo; bias; calibration | 17 |
| | 4.2 Enhanced / corrected artefacts | Outputs improving other models (predictions, data, signals) | corrected predictions; synthetic/augmented data; anomaly/OOD scores; attribution indicators | 10 |
| | 4.3 Compressed models | Quantised/tuned checkpoints with reduced cost | efficient, compressed, fine-tuned foundation models | 8 |
| **5. Embedding-space outputs** | | | | |
| | 5.1 Aligned multimodal embeddings | Joint space for text/image/audio enabling retrieval or classification | aligned multimodal embeddings; zero-shot classification | 12 |

Table 14: Structured summary of output types used in foundation-model papers.

| Category | Sub-category | What is covered | Typical examples | Cluster |
|---|---|---|---|---|
| **1. Language-modeling objectives** | | | | |
| | Next / Masked-token Prediction | Minimize CE on next/masked token. | Next-/masked-token pred.; LM; CE/NLL min.; aux reg. | 4 |
| | General LLM Advancement | Improve reasoning, alignment, efficiency, robustness. | Reasoning; alignment; eval.; efficiency; robustness; multi-domain | 12 |
| **2. Alignment & safety objectives** | | | | |
| | Human-Preference Alignment | Maximize learned reward; limit divergence. | Pref. align; reward max.; safety-divergence reg. | 1 |
| | Hallucination & Bias Mitigation | Cut hallucinations/bias via grounding alignment. | Hallucination det./mit.; x-modal align/ground; bias red. | 0 |
| | General Safety & Robustness | Losses for safety, explainability, robust autonomy. | Alignment; safety; efficiency; general.; explain.; autonomy | 6 |
| | Security & Privacy Defense | Defend attacks, watermark, erase concepts. | Adv. robustness; watermark; backdoor/membership defense; privacy; concept erase; interp. | 7 |
| **3. Adaptation & continual-learning** | | | | |
| | Prompt / Self-training Adaptation | Prompt/pseudo-label adapt for zero/few-shot. | FM adapt; prompt/pseudo-label; zero-/few-shot OVR; robustness; domain gen. | 10 |
| | Retention-Regularized Fine-tuning | Regularize fine-tuning to retain knowledge. | Task loss + retention reg.; preserve knowledge; generalization | 17 |
| **4. Multimodal objectives** | | | | |
| | Unified Multimodal Representations | Vision-language align, ground, reason. | Unif. multimodal; V-L align; grounding; x-modal reason.; zero/few-shot; cont. adapt. | 3 |
| | Contrastive & Masked Alignment | Contrastive+masked for joint embeddings. | X-modal contrast; masked recon.; joint class.; dist. align | 13 |
| | 3D / Multi-view Generation | Cross-modal loss for 3D-consistent views. | Hi-fid 3D multi-view gen.; sparse 2D/text | 19 |
| **5. Generative diffusion objectives** | | | | |
| | Core Enhancement | Faster, higher-quality diffusion via guidance. | Accelerate train/inf.; guide/loss opt.; fidelity; diversity; control | 16 |
| | Noise-prediction & Score-matching | Train via noise pred., reconstr., ELBO. | Noise pred denoise; recon. fid. min.; score/ELBO opt. | 18 |
| | Video / Motion Diffusion | Conditioned diffusion for coherent video. | Hi-fid coherent video/motion synth.; control; prompt align | 2 |
| | Controllable Image Diffusion | Steer image diffusion for fairness etc. | Align; personalise; fairness; diversity; spatial; hi fid.; light train | 5 |
| | Latent & Denoising Regularization | Extra denoise/latent loss. | Denoise min.; latent align; cond. reg.; dist. fid. train | 8 |
| **6. Policy-learning & RL** | | | | |
| | Multi-task Policy RL | One policy via cloning+pref. RL. | Multi-task policy; behavior/diffusion cloning; pref.-aligned RL; reward exp. | 14 |
| **7. Optimization & efficiency** | | | | |
| | Loss & Representation Matching | Minimize task loss, align distributions. | Task combined losses; dist align; repr match; reg. opt. | 11, 15 |
| | Compute / Memory Efficiency | Cut compute/memory, keep accuracy. | Min. compute/memory/param cost; train/ft/inf. | 9 |

Table 15: Structured summary of learning objectives used in foundation-model papers.

| Category | Sub-category | What is covered | Typical examples | Cluster |
|---|---|---|---|---|
| **1. Pre-Training & Representation Learning** | | | | |
| | 1.1 Contrastive/masked vision–language pre-training | Learn aligned image–text embeddings before any task-specific tuning. | ViT/CLIP contrastive/masked pretrain; strong aug.; $T=0.07$; 40–600ep finetune. | 5 |
| | 1.2 Adapter-aided diffusion image/video pre-training | Freeze released checkpoints; add lightweight adapters to scale. | Frozen ckpt + LoRA/prompt; AdamW + cos LR; prog-res; CF guidance. | 13 |
| **2. Fine-Tuning & Adaptation** | | | | |
| | 2.1 Vision–language instruction tuning | Turn a frozen VLM into an instruction follower. | Image–text pretrain→inst. tune; PEFT; opt. RLHF. | 1 |
| | 2.2 Parameter-efficient domain adaptation | Keep backbone frozen; adapt via prompts/adapters only. | Prompt/adapter/LoRA; distill or contrastive shift. | 10 |
| | 2.3 Instruction SFT + retrieval alignment | Align an LLM with retrieval and preferences. | Multi-stage SFT; retrieval ctx; DPO/RLHF; rerank→generate. | 6 |
| | 2.4 3-D coarse-to-fine diffusion adaptation | Make diffusion/LLM backbones 3-D consistent. | Alt. SDS/guidance; synth views; render-denoise distill. | 4 |
| | 2.5 Video-diffusion adapter tuning | Specialise image diffusion for temporal output. | Temp/spatial adapters; latent denoise; low→high-res. | 7 |
| | 2.6 Controllable diffusion sampling | Add style/identity knobs without retraining core model. | Var-score-recon losses; dyn. guidance; feature mod. | 8 |
| | 2.7 Layout / prompt-conditioned diffusion | Condition generation on structured layouts or text. | LLM layout cond.; masked-attn sampling; coarse→fine. | 11 |
| | 2.8 Composite-loss self-supervised fine-tuning | Improve a backbone with multiple unsupervised signals. | Mask/noise; contrast+recon+distill; EMA teacher. | 15 |
| | 2.9 Pseudo-label self-training | Self-train using synthetic multimodal labels. | Synth labels (Diff/LLM/SAM); filter; adapter FT; contrast/distill. | 16 |
| **3. Reinforcement Learning & Control** | | | | |
| | 3.1 Diffusion-backed policy optimisation | Blend BC and RL signals for policy training. | Traj samp; BC+PPO; Q-guided denoise; self-play. | 0 |
| | 3.2 Hierarchical planning & embodied control | Combine VLM/LLM skills with robotic policies. | Skill seg; hier plan; RH control; real-time accel. | 19 |
| **4. Efficiency & Compression** | | | | |
| | 4.1 Model compression & quantisation | Shrink models with minimal retraining. | Low-rank+sparse; mixed-prec.; prune+search. | 2 |
| | 4.2 Transformer training / inference acceleration | Architectural and parallel tricks to cut runtime. | Multi-dev partition; sparse/flash attn; KV prune; stride denoise. | 9 |
| | 4.3 Hyper-parameter & infrastructure optimisation | Well-tuned schedules and distributed stacks. | AdamW warm-cos LR; FP16/BF16; DeepSpeed; 100k–500k steps. | 17 |
| **5. Safety & Adversarial Robustness** | | | | |
| | 7.1 Jailbreak & adversarial prompt synthesis | Craft inputs that bypass safety guards. | Harmful data; shadow model; grad token opt; synth prompt. | 12 |

Table 16: Structured summary of training recipes used in foundation-model papers.

| Category | Sub-category | What is covered | Typical examples | Cluster |
|---|---|---|---|---|
| **1. Vision & Imaging Sensors** | | | | |
| | 1.1 RGB cameras | Monocular, stereo, multi-view, surround-view or panoramic cameras producing color frames; used for appearance-based perception. | front/side/rear vehicle cameras, egocentric/wrist/-head cameras, aerial/on-board cameras | 0, 1, 2, 3, 4, 6, 7, 9, 10, 11, 13, 14, 15, 16, 17, 18, 19 |
| | 1.2 RGB-D cameras | Active or structured-light cameras that output synchronized color + depth images. | Intel RealSense, Azure Kinect, panoramic RGB-D rigs | 3, 7, 10, 11, 13, 14, 17, 18, 19 |
| | 1.3 Event (neuromorphic) cameras | Asynchronous sensors emitting per-pixel brightness changes with micro-second latency. | DVS, DAVIS | 9 |
| | 1.4 Thermal / LWIR cameras | Passive long-wave IR imagers for temperature or night-vision cues. | Thermal cameras, LWIR DoFP polarization cameras | 3, 14 |
| **2. Depth & Range Sensors** | | | | |
| | 2.1 LiDAR | Spinning or solid-state laser scanners returning 3-D point-clouds. | Multi-beam/spinning LiDAR, PolLidar wavefront lidar | 3, 4, 5, 11, 13, 14, 16, 17 |
| | 2.2 Time-of-Flight cameras | Pulsed or continuous-wave light cameras computing per-pixel range. | Indirect/monocular ToF depth cameras, AMCW-ToF | 9, 14 |
| | 2.3 Radar | mmWave / FMCW / 4-D imaging radars measuring range–Doppler or heat-maps. | Automotive mmWave/FMCW, MIMO imaging radar | 3, 4, 14 |
| **3. Proprioceptive Sensors** | | | | |
| | 3.1 Joint & wheel encoders | Optical or magnetic sensors giving joint angle / wheel ticks. | joint encoders, wheel encoders | 3, 7, 8, 13, 16 |
| | 3.2 IMUs | 3-axis accelerometers & gyros providing orientation/velocity. | IMU, pose modules | 3, 4, 7, 13, 16, 17 |
| | 3.3 Force / torque sensors | Strain-gauge or multi-axis transducers measuring interaction forces. | force–torque sensors, motor-current feedback | 7, 13, 16, 19 |
| | 3.4 Motor-current sensors | Drive-current read-back for inferred load. | motor-current feedback | 19 |
| **4. Tactile & Contact Sensors** | | | | |
| | 4.1 Vision-based tactile | Camera-in-gel sensors capturing high-resolution surface contact. | GelSight, Soft-Bubble | 13 |
| | 4.2 Pressure / tactile arrays | Capacitive or resistive skins giving per-taxel pressure maps. | force-torque/pressure arrays, contact sensors | 7, 13 |
| **5. External Tracking & Global Localization** | | | | |
| | 5.1 Optical motion-capture systems | Infra-red camera networks tracking reflective markers. | VICON, optical marker rigs | 3, 13, 14, 19 |
| | 5.2 Wearable mocap devices | Marker gloves or body suits for fine human-pose capture. | motion-capture gloves, skeletal/hand markers | 13, 19 |
| | 5.3 Radio-based positioning | Satellite or UWB transceivers returning global coordinates. | GPS, UWB beacons | 3, 4, 16 |
| **6. Audio Sensors** | | | | |
| | 6.1 Microphones / audio arrays | Mono or array microphones for speech / environmental sound. | microphone audio inputs | 3, 13, 19 |

Table 17: Structured summary of input sensors used in robotic papers.

| Category | Sub-category | What is covered | Typical examples | Cluster |
|---|---|---|---|---|
| **1. Ground-based mobile robots** | | | | |
| | 1.1 Small RC / off-road vehicles | 1/10-scale cars, ATVs, skid-steer rovers for field tests | RC cars/ATVs; small off-road vehicles | 19 |
| | 1.2 Kinematic vehicle models | Bicycle/unicycle point-mass models (simulation-only) | Simulated vehicle agents (kinematic/dynamic) | 0 |
| **2. Aerial robots** | | | | |
| | 2.1 Quadrotors / drones | Four-rotor UAVs with cameras, LiDAR, IMU | Quadrotor UAVs; drones | 11, 19 |
| **3. Legged & humanoid robots** | | | | |
| | 3.1 Simulated legged agents | Classic MuJoCo bodies for RL locomotion | Hopper; HalfCheetah; Walker2d; Ant; Quadruped | 11 |
| | 3.2 Real quadrupeds & hybrids | Torque-controlled ∼12-DoF quadrupeds; wheel-leg hybrids | ANYmal; Unitree A1/Go1; MIT Mini-Cheetah; wheel-leg hybrids | 12 |
| | 3.3 Humanoids | High-DoF bipeds/humanoids, often with articulated hands | Humanoids/bipeds; SMPL-X mesh; simulated avatars | 16 |
| **4. Manipulators & end-effectors** | | | | |
| | 4.1 Standard 6–7 DoF arms | Fixed-base arms with two-finger or suction grippers | UR5e; Sawyer; other 6–7 DoF arms | 2 |
| | 4.2 Franka-class agile arms | 7-DoF Panda-style arms popular in RL/IL | Franka Emika Panda; Robotiq; suction cups | 3 |
| | 4.3 Mobile / dual-arm manipulators | One or two arms on a wheeled base (bimanual possible) | Mobile bases with dual arms; mobile manipulators | 7, 11 |
| | 4.4 Arm + dexterous hand | Arms distinguished by multi-finger hands | Shadow; Allegro; Adroit; LEAP; DeltaHand | 16, 18 |
| **5. Soft & continuum robots** | | | | |
| | 5.1 Continuum / soft manipulators | Deformable backbones, pneumatic/tendon actuation, soft skins & grippers; tensegrity frames | Soft continuum arm; soft gripper; compliant tensegrity structures | 1 |

Table 18: Structured summary of physical bodies used in robotic papers.

| Category | Sub-category | What is covered | Typical examples | Cluster |
|---|---|---|---|---|
| **1. Direct joint-level outputs** | | | | |
| | 1.1 Joint state read-outs | Instantaneous articulated joint positions, orientations, angles, velocities | "joint positions & orientations"; "joint angles"; "joint velocities"; "mesh deformations" | 0, 1, 9 |
| | 1.2 Joint command signals | Low-level motor targets (torque / position / velocity) that drive joint motion | "joint torque/position commands"; "continuous motor control signals"; "PD control torques" | 7, 11, 12, 16, 17 |
| | 1.3 Joint motion trajectories | Time-indexed sequences of joint states the robot follows | "motion sequences over time"; "planned joint trajectories"; "optimised 6-DoF trajectories" | 0, 1, 17 |
| **2. Rigid-body / end-effector pose outputs** | | | | |
| | 2.1 6-DoF body poses | Position + orientation of whole robots, cameras or objects | "6-DoF poses"; "rigid transformations"; "UAV 3-D position & orientation" | 6, 10, 16 |
| | 2.2 End-effector pose + gripper | Cartesian pose of manipulator tip plus gripper open/close state | "6-DoF end-effector pose $(x, y, z, r, p, y)$"; "gripper_state (open/close)" | 4 |
| **3. Ground-vehicle / mobile-robot control outputs** | | | | |
| | 3.1 Steering & pedal commands | Low-level automotive controls for heading and speed | "steering_angle"; "acceleration/throttle"; "brake" | 3 |
| | 3.2 Wheel / differential-drive velocities | Body-frame linear & angular velocity commands for wheels/actuators | "linear & angular velocity motor commands"; "wheel/actuator motions" | 14 |
| | 3.3 Motion trajectories | Pre-planned paths or waypoints for vehicle motion | "robot/vehicle motion trajectories"; "position/orientation updates" | 19, 16 |
| **4. Aerial-rotorcraft control outputs** | | | | |
| | 4.1 Rotor thrust & body-rate commands | Per-rotor thrust/speed or body-rate inputs that place a UAV in 3-D space | "rotor thrust/speed commands"; "collective thrust & body-rate inputs" | 6 |

Table 19: Structured summary of joint outputs used in robotic papers.

| Category | Sub-category | What is covered | Typical examples | Cluster |
|---|---|---|---|---|
| **1. Continuous Low-Level Actuation** | | | | |
| | 1.1 Joint-space commands | Direct numerical inputs to individual joints or actuators, bounded by hardware limits. | joint torques/positions/velocities; high-dimensional joint commands; bounded control inputs; finger-joint configs; parametrised joint trajectories | 0, 4, 6, 10, 12, 14, 18 |
| | 1.2 Vehicle / body dynamics commands | Low-level controls that change a mobile base, ground-vehicle or aerial body state. | steering angle; throttle / acceleration; braking; linear & angular velocity; body-rate thrust; speed/direction for locomotion; lane-keeping | 0, 1, 7, 10, 12, 13, 15 |
| **2. Mid-Level Pose & Trajectory Control** | | | | |
| | 2.1 End-effector & gripper pose | 6-DoF goals and time-parameterised trajectories for arms, grippers or aerial manipulators. | continuous 6-DoF poses; pose deltas ($\Delta x, \Delta y, \Delta z, \Delta$roll, $\Delta$pitch, $\Delta$yaw); gripper open/close; gripper width/force; grasp trajectories | 2, 6, 9, 10, 12, 14, 18 |
| | 2.2 Base / waypoint trajectories | Desired paths, way-points or velocity profiles for the robot body or ego vehicle. | waypoint/path-goal selection; future trajectory sequences; base linear & angular velocity commands; lane-change/merge trajectories | 0, 1, 7, 10, 15, 19 |
| **3. High-Level Discrete Skills & Behaviour Primitives** | | | | |
| | 3.1 Manipulation skills | Object-centred primitives that parameterise targets, forces or object states. | grasp/pick; place/drop; push/pull; rotate/open/close; part deformation | 0, 10, 18, 19 |
| | 3.2 Locomotion & navigation skills | Discrete moves or gait switches for repositioning the whole robot. | move_forward/stop; turn_left/turn_right; gait switch; lane keeping/change; overtaking/merging; "go to X" | 0, 1, 10, 15, 19 |
| | 3.3 Interaction & instruction skills | Multimodal actions expressed through gesture, speech or scene edits. | gesture actions; speech actions; instructional guidance; scene editing commands | 0 |

Table 20: Structured summary of action space used in robotic papers.

| Category | Sub-category | What is covered | Typical examples | Cluster |
|---|---|---|---|---|
| **1. Autonomous-driving & Mobile-vehicle scenes** | | | | |
| | 1.1 On-road urban / suburban / rural driving | Real or simulated road networks with traffic, road rules, and weather variation. | urban roads; highways; intersections; traffic lights/signs; . . . | 1, 2, 6, 9, 12, 13, 19 |
| | 1.2 Off-road, cross-country & planetary terrain | Structured or unstructured natural terrains requiring ground-robot locomotion. | uneven ground; sand; gravel; snow; . . . | 11 |
| **2. Manipulation workspaces** | | | | |
| | 2.1 Basic household tabletop | Small cluttered indoor bench for reach-scale manipulation. | cluttered tabletop; household objects; articulated fixtures; . . . | 0 |
| | 2.2 Kitchen & household benchmark suites | Standardised kitchen/tabletop scenes from RLBench, Meta-World, FrankaKitchen, Habitat, Ravens, etc. | kitchen counters; RLBench station; FrankaKitchen; . . . | 14, 17 |
| | 2.3 Assembly & insertion tables | Contact-rich assembly surfaces with precisely shaped parts. | assembly workspace; peg-hole joints; plug-socket joints; . . . | 18 |
| | 2.4 Shared lab / industrial workcells | Planar or 3-D manipulation bays in labs or factories, often human-robot shared. | lab work surfaces; human-robot zones; static & dynamic obstacles; . . . | 10 |
| **3. Embodied navigation & Scene-understanding worlds** | | | | |
| | 3.1 Multi-room home / office interiors | Photorealistic or simulated domestic & office floorplans for navigation and light manipulation. | apartments; offices; corridors; dynamic changes; . . . | 7 |
| | 3.2 Large-scale mixed indoor-outdoor simulators | Dynamic 3-D worlds with physics for point-goal, exploration, or social-navigation tasks. | rooms; mazes; multiple agents; partial observability; . . . | 15 |
| | 3.3 Object-rich mixed-reality scene sets | Real + synthetic household, lab, or industrial spaces emphasising clutter & diversity. | household rooms; industrial floors; cluttered indoor scenes; . . . | 4, 5, 8 |
| **4. Physics-centric control benchmarks** | | | | |
| | 4.1 Classic locomotion & manipulation suites | Widely-used control benchmarks with domain-randomised dynamics. | MuJoCo tasks; IsaacGym walkers; Robotarium arena; . . . | 3 |
| | 4.2 High-fidelity multi-physics platforms | Environments that model contact, fluids, deformables & human interaction in indoor/outdoor scenes. | rigid-body scenes; deformable objects; fluid interaction; humans; . . . | 16 |

Table 21: Structured summary of environment used in robotic papers.

# D  TREND ANALYSIS

Figure 5: Trend Visualization of Computer Vision Research

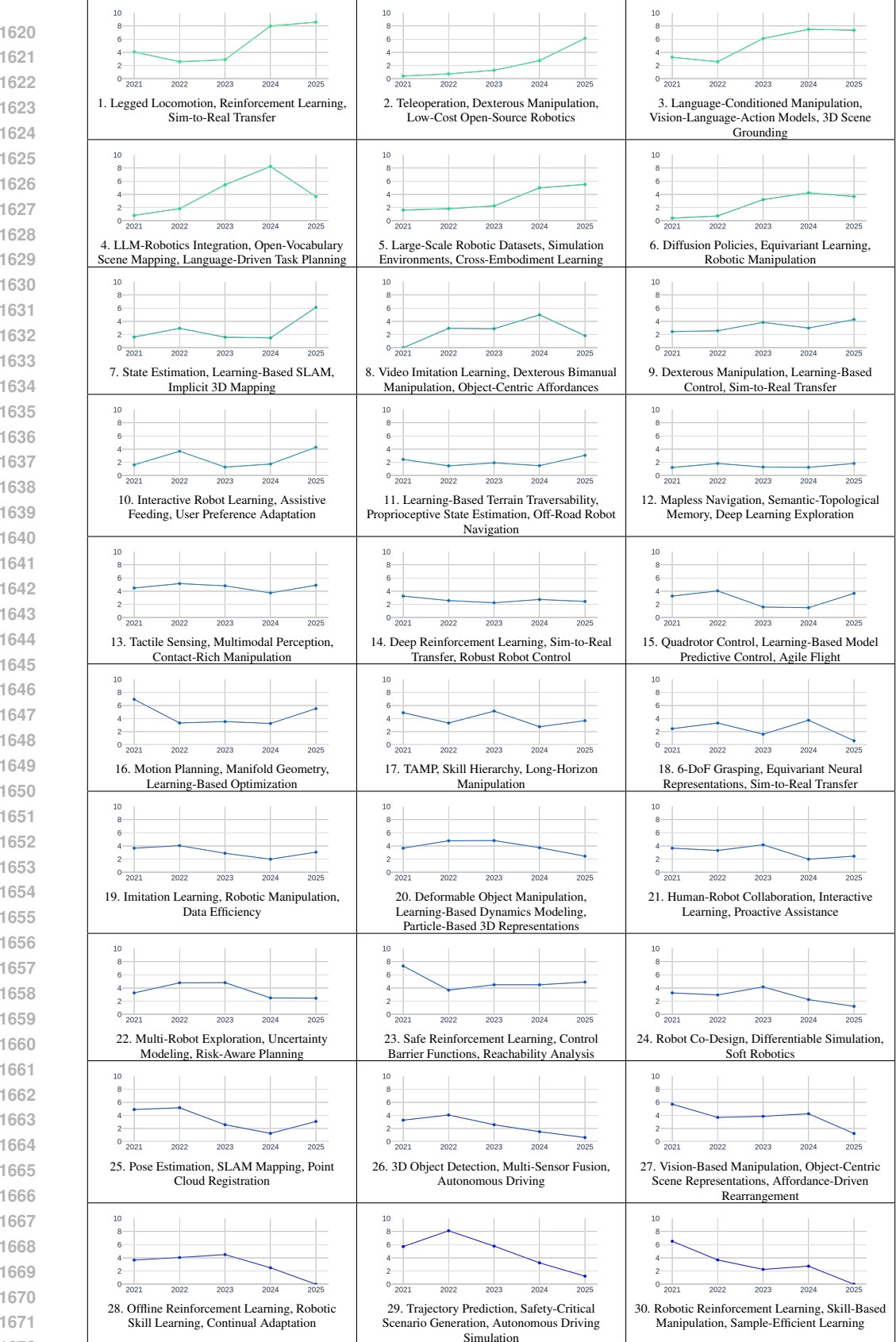

Figure 6: Trend Visualization of Robotics Research

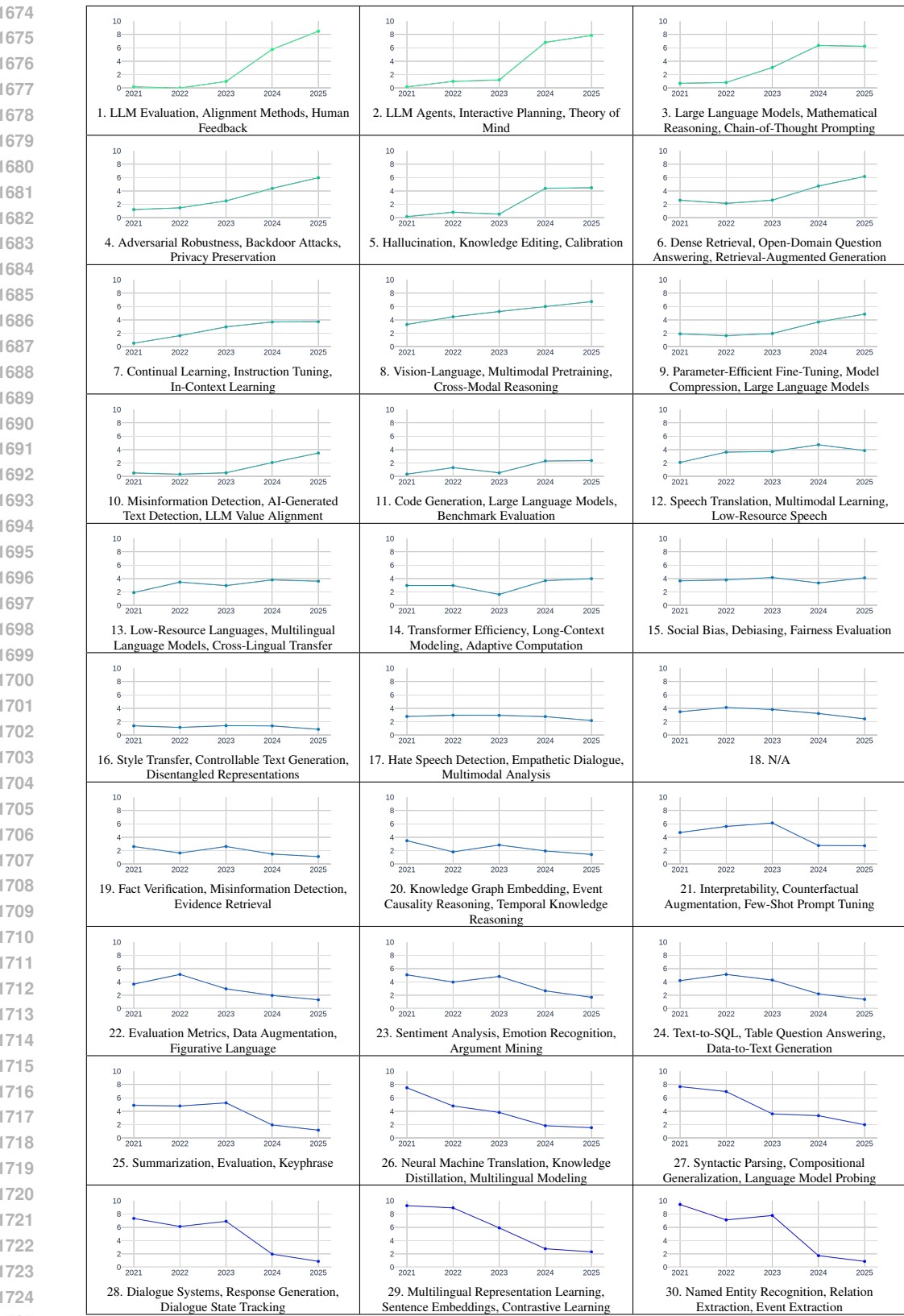

Figure 7: Trend Visualization of NLP Research

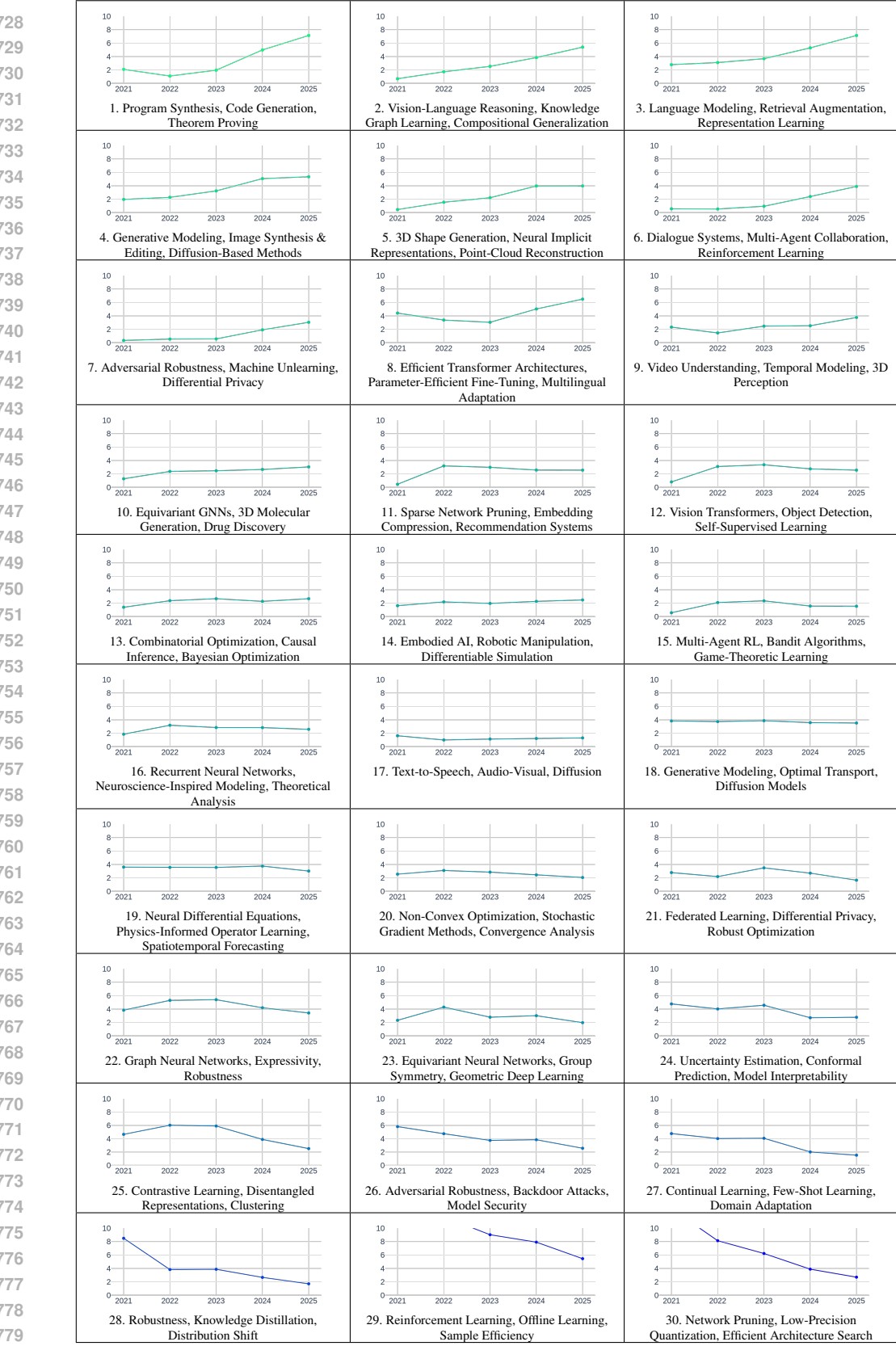

Figure 8: Trend Visualization of Machine Learning Research

