# OpenReview forum: "Real Deep Research from Foundation Model to Robotics"
_ICLR.cc/2026/Conference — Submitted to ICLR 2026_

### Official Review · Reviewer_33WT · 2025-10-21

**Soundness:** 3
**Presentation:** 3
**Contribution:** 2
**Rating:** 2
**Confidence:** 4

**Summary:**

This paper presents Real Deep Research (RDR), a framework designed to automate large-scale literature analysis and trend discovery. The motivation is the increasing difficulty for researchers to stay up to date, given the exponential expansion of ML papers. The proposed RDR pipeline integrates large language models (LLMs) and embedding-based analysis through four stages: (1) topic filtering, where papers are selected based on relevance to defined research areas; (2) content structuring, where LLM extracts key information using expert-defined perspectives (e.g., input, modeling, output); (3) projection, mapping textual content into a embedding space; and (4) embedding analysis, clustering papers to identify themes and trends. The system outputs structured surveys, trend visualizations, and cross-domain knowledge maps. The authors benchmark RDR against commercial LLM tools (e.g., GPT-5, Gemini) and report higher survey quality based on pairwise human preference evaluations.

**Strengths:**

1. originality: The paper presents an original attempt towards automatic meta-research using a combined approach of LLM-based reasoning, embedding projection, and clustering.
2. quality: The paper clearly articulates the modular stages of the RDR pipeline and conduct both qualitative and quantitative evaluations.
3. clarity: The four-stage pipeline is easy to follow.
4. Significance: The problem addressed is timely and significant, as ML paper output grows exponentially, automating meta-research is necessary.

**Weaknesses:**

1. Problem formulation and evaluation criteria are vague: while the paper sets out to help researchers stay up to date with the rapidly growing literature, the problem is framed too vaguely and lacks precise, measurable objectives. The claimed outcomes (e.g., “uncovering cross-domain opportunities,” and “offering concrete starting points”) are not defined as quantitative metrics. For example, the paper claimed “identifying emerging trends,”. How much improvement is achieved with RDR compared to a simple alternative, such as keyword-frequency analysis?  Could we measure practical impact via recall of relevant papers or novelty of suggested research direction?

2. Interdisciplinary claims not substantiated: Although the authors emphasize interdisciplinary exploration as a major goal, there is a lack of evidence for how RDR successfully discovers non-trivial cross-domain connections that inspire new research. The “knowledge graph” analysis is mentioned but not described in detail. Demonstrating even one verified interdisciplinary novel insight, e.g., validated by experts, would greatly strengthen the paper.

3. ablation study without embedding model and the content reasoning step? Why are the perspectives useful?

**Questions:**

1. The “area filtering” stage relies on manually crafted LLM prompts to decide which papers qualify as belonging to “foundation model” or “robotics” domains. Could this step imposes rigid domain boundaries that contradict the paper’s stated aim of enabling interdisciplinary exploration?

2. The introduction of symbolic notation and equations such as F(p) and D(p) adds complexity without clear benefit.

3. How consistent were human evaluators in the survey-quality study? Any in-depth analysis that could help clarify how the embedding stage contributes to improved survey quality?

4. Some tables and figures lack captions. For the table in lines 324-331, each row appears to be associated with multiple clusters. How are they identified?

5. typo in line 238, should be LLM?

6. Could make clear of the underlyinig assumption with the clustering analysis. For example, is it assumed that there are k independent and well-separated concepts for each field? Any hirerchical / overlapping examples you noticed?

---

### Official Review · Reviewer_aKLT · 2025-10-25

**Soundness:** 3
**Presentation:** 3
**Contribution:** 3
**Rating:** 4
**Confidence:** 4

**Summary:**

The paper introduces Real Deep Research, an automated, LLM-based pipeline for systematically surveying research areas, identifying emerging trends, and discovering interdisciplinary opportunities. Applied primarily to AI and robotics (with extensions to other sciences), the framework combines paper crawling, LLM-based filtering, embedding projection, and clustering-based trend analysis. It claims to outperform commercial LLMs in generating structured, accurate surveys and in unsupervised clustering benchmarks, arguing that RDR offers a scalable and generalizable alternative to manual surveys.

**Strengths:**

1. Well articulated motivation (addressing information overload in AI/robotics research with automation) and comparison with related work.
2. Clear modular pipeline design (data collection, reasoning, projection, analysis) with the potential to customize perspectives for different domains.
3. The paper is well presented, with clear figures and tables.

**Weaknesses:**

1. Evaluation is not convincing. The paper does not explain the human study design, like number and background of participants, recruitment process, evaluation questions (scoring criteria/metrics), questions and input shown to them. Please clarify these details in the rebuttal. I am willing to raise my score if the evaluation is sound.
2. No ablation or sensitivity study on pipeline components (e.g., prompt choices, embedding models).
3. Minor: It might also be worth exploring whether this structured embedding could facilitate novel idea generation, particularly across multidisciplinary domains.

**Questions:**

1. Could the authors clarify the human study design in more detail, including the number and background of participants, recruitment process, evaluation questions, scoring metrics, and the exact inputs shown to participants?
2. Have the authors conducted any ablation or sensitivity analysis on the pipeline components?

---

### Official Review · Reviewer_sowZ · 2025-10-30

**Soundness:** 3
**Presentation:** 2
**Contribution:** 2
**Rating:** 2
**Confidence:** 3

**Summary:**

The paper proposes a pipeline to analyze (large) quantities of research papers with regards to emerging trends, cross-domain opportunities and concise questions/starting points to investigate.

**Strengths:**

* The problem is timely and important
* The empirical results (Table 2) appear very compelling, though with limitations (see weaknesses)
* The problem is complex and paper fairly comprehensive

**Weaknesses:**

* The paper is solid engineering, but without deeper contributions or novel aspects.
* The empirical results need more details: It is not clear how many domain experts participated also the demographics of them, where they were recruited, if randomization took place etc.
* The results lack any form confidence intervals or statistical testing, though overall gains seem huge, but there are concerns that only very few tests have been performed (see questions). Also it is not clear if the evaluation was fair, i.e., the effort for engineering other approaches such as prompt engineering for GPT5-Research was adequate.
* Also the comparison with off the shelf LLMs might not be best, as there are a number of papers focusing on summarizing papers and even writing papers (generating ideas).
* The choice of papers is ok, but not optimal for trends (see also questions)
* The amount of papers is relatively small. Table 1 shows more than 30k papers but they effectively only use about 5500 as written in the text - this year on AAAI alone there were 28k submissions... (The Table 1 is rather confusing. )


Detail: abstract filed -> field
Appendix. B. -> Appendix B. (also other places e.g. for C/D
The table refs seem incorrect, e.g. "As shown in Tab. 5, our method, Real Deep
Research (RDR), achieves the highest overall performance with an average rank of 1.30,"   -> Table 2?
"As shown in Tab. 5, our method RDR achieves the best performance across both datasets, with an accuracy of 84.86 " -> Table 3 ?

**Questions:**

*  Why did you not include arxiv.org? If you want to get trends, you lag one year behind if you wait for proceedings, which is a long time.
  "In total, we collected 8 evaluation entries, each with 80 pairwise comparisons. " So there were only 8*80 pairwise assessments or is it 8*80*#experts (which is not known, or seems burried at least)

---

### Official Review · Reviewer_5aqV · 2025-11-02

**Soundness:** 2
**Presentation:** 3
**Contribution:** 1
**Rating:** 2
**Confidence:** 4

**Summary:**

The paper presents a pipeline for analyzing scientific papers in specific domains.

**Strengths:**

Deep research is an interesting and relevant topic.

The paper is well written and easy to follow.

The presentation is described in sufficient detail to have a reasonable chance to be reproducible.

**Weaknesses:**

There are no significant technical results. The paper basically describes an instance of deep research applied to foundations models and robotics.

The notation used for filtering paper is unclear, see question.

The reference to Tab 5 should be Tab 2.

**Questions:**

Why do you use an LLM for the foundation models and an LMM for the robotics papers?

Can you please explain what F(p) is? Is it a probability? Is a paper? The domain D seems to be a set of papers, thus F(p) ought to be a paper, but stating that F=LLM(p|...) seems to indicate it is a probability. Is it the probability to include it, and then you throw a dice to determine if the paper is actually included in D?

The perspectives are derived from the paper, why make the probability of p dependent on them? I guess it goes back to how to interpret the definitions of F, see above.

Since RDR is also using commercial tools, how does the choice affect the result? Table 2 compares directly asking commercial tools compared to using the RDR pipeline.

---

### Meta-Review · Area_Chair_Fn4Y · 2026-01-05

**Summary:**

The main critiques are that:
- there is lack of significant technical novelty to machine learning, and the paper is instead a presentation of an engineered system. It appears to produce useful outputs, but it is unclear what the core principles are, and how well it can be reproduced.
- evaluation is insufficiently detailed (human study design is unclear, participants, recruitment, metrics, inputs), insufficiently rigorous (baseline engineering / statistical measures etc.), and the problem definition / evaluation metrics themselves are vague.


There are also more minor gripes about the specific choices of datasets, the tension between interdisciplinarity and area filtering, writing issues and typos etc.

And finally, there are several questions about the system and experiment design that remain unaddressed in the absence of an author response.

**Reviewer Concerns:**

No author response was provided, so all concerns are outstanding.

**Reviewer Scores:**

Unchanged.

---

### Decision · Program_Chairs · 2026-01-26

Reject